# LESS IS MORE: STEALTHY AND ADAPTIVE CLEAN-IMAGE BACKDOOR ATTACKS WITH FEW POISONED

## ABSTRACT

Deep neural networks are fundamental in security-critical applications such as facial recognition, autonomous driving, and medical diagnostics, yet they are vulnerable to backdoor attacks. Clean-image backdoor attack, a stealthy attack utilizing solely label manipulation to implant backdoors, renders models vulnerable to exploitation by malicious labelers. However, existing clean-image backdoor attacks likely lead to a noticeable drop in Clean Accuracy (CA), decreasing their stealthiness. In this paper, we show that clean-image backdoor attacks can achieve a negligible decrease in CA by poisoning only a few samples while still maintaining a high attack success rate. We introduce **G**enerative Adversarial **C**lean-Image **B**ackdoors (GCB), a novel attack method that minimizes the drop in CA to less than 1% by optimizing the trigger pattern for easier learning by the victim model. Leveraging a variant of InfoGAN, we ensure that the trigger pattern we used has already been contained in some training images and can be easily separated from those feature patterns used for benign tasks. Our experiments demonstrate that GCB can be adapted to 5 datasets—including MNIST, CIFAR-10, CIFAR-100, GTSRB, and Tiny-ImageNet—5 different architectures, and 4 tasks, including classification, multi-label classification, regression, and segmentation. Furthermore, GCB demonstrates strong resistance to backdoor defenses, successfully evading all detection methods we know. Code: `anonymous.4open.science/r/GCB`.

## 1 INTRODUCTION

Deep Neural Networks (DNNs) are widely used in applications like facial recognition (An et al., 2023), autonomous driving (Han et al., 2022), and medical image diagnosis (Li et al., 2021a); however, backdoor attacks threaten their trustworthiness. By poisoning a small portion of the training data (Li et al., 2022), adversaries can inject backdoors that cause models to make erroneous predictions when specific inputs are presented. Recent studies reveal that backdoors can be implemented without modifying images, known as *clean-image backdoors*—a significant concern when data annotation is outsourced to third parties. For instance, Chen et al. (2022a) induced a one-to-one backdoor attack in multi-label classification by relabeling images from a source label to a target label, though this method is less adaptable to general image classification tasks. To address this limitation, Jha et al. (2024) proposed a label-optimization technique that constructs a surrogate poisoned-image backdoor model and optimizes soft labels to mimic its behavior.

Although these methods achieve a high Attack Success Rate (ASR), they experience a significant drop in Clean Accuracy (CA), limiting their stealthiness in practice. For instance, the state-of-the-art method FLIP (Jha et al., 2024) shows a CA reduction of 1.7% and a poison rate of 2% when averaged across all classes in a one-to-one scenario (one source and one target class). However, a closer analysis reveals that the CA drop and poison rate are much more pronounced at the class level. As shown in Fig. 1, 15.8% of inputs from the source class were poisoned, resulting in a 13.3% CA drop for that class. Even in the all-to-one scenario, FLIP leads to a consistent CA drop of 4.0% across all classes. This significant drop should alert the victim to a potential backdoor when examining the accuracy of each class on a clean test set.

The significant drop in CA can be attributed to a phenomenon known as the natural backdoor trigger in clean-image backdoor, first introduced in (Rong et al., 2024). When a small percentage (e.g., 5%) of training images are relabeled to train a poisoned victim model, the i.i.d. properties of the training

and test datasets result in approximately the same proportion of testing images (around 5% in this example) being misclassified by the poisoned victim model. This leads to a substantial drop of about 5% in CA. Unfortunately, this effect is applicable to all types of clean-image backdoors, regardless of the specific attack methods employed. This raises a critical question: *Can we mitigate this effect to create a more stealthy clean-image backdoor?*

Our answer is affirmative. By proposing Generative Adversarial Clean-Image Backdoors (GCB), we can significantly reduce the poison rate to 0.1%. This leads to substantial mitigation of the CA drop, averaging only 0.2% across classes and a maximum of 0.5% for any single class. Our key idea is to lower the poison rate by optimizing the trigger pattern, making it easier for the victim model to learn. However, within the context of clean-image backdoors, optimizing the trigger pattern is challenging because we cannot modify images; instead, we must utilize features that already exist on benign images to construct triggers. In this context, three main constraints arise: **(1) Existence**: The optimized trigger pattern must be present in the training set. **(2) Separability**: Images with and without the trigger must be easily distinguishable, enabling the victim model to learn the backdoor more effectively and low poison rate. **(3) Irrelevancy**: The trigger should not interfere with learning the benign task, preventing a significant CA drop.

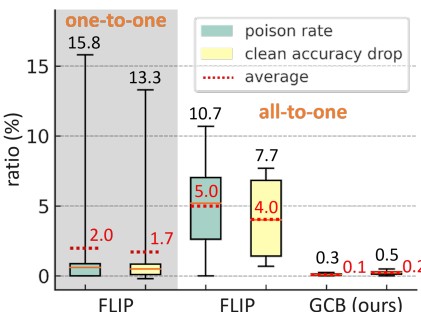

Figure 1: Box plot comparing clean-image backdoors across all classes on CIFAR-10. The SOTA method, FLIP, has a low average clean accuracy drop (1.7%) but can reach up to 13.3% in maximum. In contrast, our method shows almost no clean accuracy drop.

In this paper, we develop a novel GAN framework, C-InfoGAN, to optimize triggers while addressing three key issues. (a) To ensure *existence*, we employ a GAN generator to construct a trigger function, ensuring that all generated images, including trigger images, belong to the original image distribution. (b) To achieve *separability*, we adopt the concept of InfoGAN, building the GAN generator in a two-fold manner (representing triggered and benign images respectively) and maximizing their distance as a term in the loss function. (c) To guarantee *Irrelevancy*, we incorporate the ground truth label as a prior for all components to ensure that the trigger features are irrelevant to class features.

We conducted extensive experiments to validate the effectiveness of GCB. GCB achieved impressive ASRs of 97.9%, 100%, 92.1%, and 94.1% with only 0.5% data poisoning per dataset, while maintaining less than a 1% drop in clean accuracy. Remarkably, GCB extends to various supervised vision tasks like multi-label classification, regression, and segmentation. Furthermore, when adversaries can access only 10% of the total dataset and poison 1% within that portion, GCB still attains a 90.3% ASR with only a 0.15% drop in clean accuracy on CIFAR-10. Additionally, GCB demonstrates robustness against comprehensive backdoor defense and mitigation strategies. In addition, unlike previous methods that struggle to scale to large datasets or transfer to advanced architectures, our approach performs well across various datasets—including MNIST, CIFAR-10, CIFAR-100, GTSRB, and Tiny-ImageNet—and supports different model architectures, such as ResNet, VGG, and ViT.

Our contributions are three-folded. ❶ **Outstanding stealthiness:** Our GCB can introduce clean-image backdoors with tiny poison rate ($\leq 1\%$) and minimal CA drop ($\leq 1\%$) to achieve ASR over 90% on all tested datasets. ❷ **Strong Adaptivity:** Our method adapts to 5 datasets of different scales, 5 different architectures, and 4 different tasks. ❸ **Novel Attack Method:** We innovatively design a variant of InfoGAN, C-InfoGAN, to solve the trigger optimization problem, which makes the trigger easier to learn without interfering with benign task learning.

## 2 RELATED WORK

### 2.1 DATA POISONING BACKDOOR

This paper introduces GCB, a novel backdoor attack via data poisoning. Previous attacks have evolved over time. BadNets (Gu et al., 2019) pioneered backdoor attacks in DNN models by injecting a small set of trigger-embedded data into the training set, causing misclassification when triggers appear during testing. Chen et al. (2017) advanced this concept using blending strategies to generate

poisoned images, making triggers less visible but still detectable by defenders. Liu et al. (2020) employed natural reflections as triggers, leveraging common physical phenomena to mask the attack, thereby enhancing its real-world applicability. Turner et al. (2019) introduced clean-label backdoor attacks, which perturb input images without altering labels, making malicious samples more stealthy.

Our GCB is also a clean-image backdoor attack. Clean-image backdoors were first proposed in CIB (Chen et al., 2022a), which is designed for multi-label classifications. This method relabels all images with a particular combination of labels, thus failing to generalize to standard image classification tasks and all-to-one attacks. Jha et al. (2024) proposed FLIP to implement clean-image backdoors in common classification tasks by optimizing soft labels to imitate the behaviors of poison-image backdoors. However, it heavily

Table 1: Comparison of Clean-Image Backdoors.

| Property | CIB | FLIP | CIBA | GCB(our) |
|---|---|---|---|---|
| Poison rate $\leq 1\%$ | ○ | ○ | ○ | ● |
| CA drop $\leq 1\%$ | ○ | ○ | ○ | ● |
| ASR $\geq 90\%$ | ● | ● | ○ | ● |
| Scalability | ● | ○ | ○ | ● |
| Transferability | ● | ○ | ● | ● |
| Classification | ○ | ● | ● | ● |
| Multi-label Tasks | ● | ○ | ○ | ● |

relies on matching the expert model structure with that of the victim model C.1, making it impractical in real-world situations. Additionally, due to the intuitive design of the trigger, FLIP faces significant scalability issues. It can only achieve high ASR on CIFAR-10 and CIFAR-100 with 10 coarse labels, failing to generalize to datasets with more labels. Rong et al. (2024) proposed CIBA to create an invisible clean-image backdoor by minimizing trigger perturbation. However, it can achieve less than 50% ASR even on CIFAR-10, which significantly degrades its applicability. As shown in Table 1, our method can overcome all the shortcomings mentioned in the three clean-image backdoors.

## 2.2 GAN-BASED REPRESENTATION LEARNING

In GCB, we employ C-InfoGAN, a GAN-based representation learning technique, to enhance our injected backdoor's efficiency. Significant developments have marked the evolution of GANs in interpretable and controllable representation learning. The original GAN (Goodfellow et al., 2014) established foundational frameworks but faced challenges with unstructured latent noise. Advancements like cGAN (Mirza and Osindero, 2014) incorporated label information for controlled generation, and InfoGAN (Chen et al., 2016) introduced mutual information loss to enhance interpretability. StyleGAN (Karras et al., 2019) provided nuanced control over the noise vector, enabling precise image adjustments. However, these models primarily manipulated generated images and had limited ability to edit real-world images. GAN Inversion (Xia et al., 2022) tackled this by adding a network to map real images into the GAN's latent space, increasing complexity. In our paper, we propose a novel architecture called C-InfoGAN, which integrates feature editing capability directly into the GAN architecture while ensuring interpretability of the controlled features.

## 3 PRELIMINARY

### 3.1 THREAT MODEL

We adopt the same threat model as other clean-image backdoors (Jha et al., 2024; Chen et al., 2022a): investigating the risks posed by third-party malicious annotators in the context of a large, externally annotated dataset. In this scenario, we consider attackers to have partial or full access to view the training dataset, but their malicious actions are limited to subtly mislabeling a small portion of the dataset, without the ability to modify the images or influence other training aspects like the architecture or training schedule.

### 3.2 NOTATION

In this study, we consider a supervised learning scenario for a model, $f$, defined by $y = f(x)$, where $x$ is the input and $y$ is the output label. In our GCB attack, the attacker divides the input set $X$ into benign ($X_0$) and malicious ($X_1$) subsets. The malicious subset $X_1$ is uniformly relabeled with a target label $y_t$, forming $(X_1, Y_1) = \{(x, y_t) : x \in X_1\}$. The entire dataset then becomes $(X, Y') = (X_0, Y_0) \cup (X_1, Y_1)$, where $(X_0, Y_0)$ retains the original benign labels. The cardinality of $X_1$ is constrained by the poison rate $p_r$, such that $|X_1| = p_r \cdot |X|$. The attacker's goal is to make the

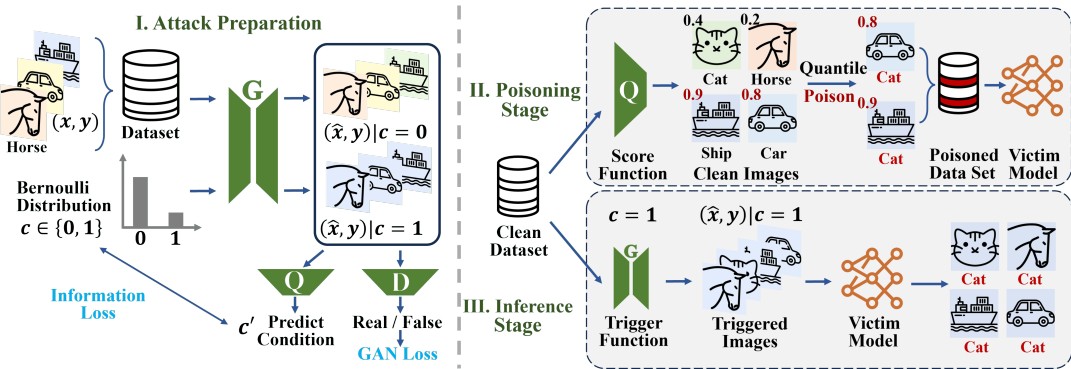

Figure 2: Framework of Generative Adversarial Clean-Image Backdoors (GCB). In the preparation stage, a specific clean feature (e.g., background color here) is extracted as a backdoor trigger.

victim model learn the following two tasks simultaneously:

$$f_\theta^* = \arg\min_\theta \underbrace{\mathbb{E}_{(x_0,y_0)\sim(X_0,Y_0)}[\ell(f_\theta(x_0),y_0)]}_{\text{classification task}} + \underbrace{\mathbb{E}_{(x_1,y_t)\sim(X_1,Y_1)}[\ell(f_\theta(x_1),y_t)]}_{\text{backdoor task}} \quad (1)$$

During testing, a trigger function $T(\cdot)$ converts benign inputs $x$ into triggered inputs $\hat{x} = T(x)$, activating the backdoor to mislead the victim model to predict the target class $y_t = f_\theta^*(T(x))$.

## 4 METHODOLOGY

### 4.1 OVERVIEW

GCB aims to minimize the CA drop while maintaining a high ASR for clean-image backdoors. In these scenarios, a portion of training images are deliberately mislabeled, but the images themselves remain unchanged. To select images to mislabel, we introduce a new network C-InfoGAN, that is trained to recognize patterns present in some training images but distinct from those patterns used for benign tasks. The GCB framework is illustrated in Fig. 2. GCB comprises three stages: attack preparation, poisoning, and inference. During attack preparation, the C-InfoGAN is trained to identify these specific patterns. Subsequently, we utilize the $Q$ component of C-InfoGAN to identify training images with the pattern and mislabel them. In the inference stage, we use the $G$ component to convert any image into a triggered input, misleading the victim model to predict $y_t$.

### 4.2 C-INFOGAN

Essentially, given a fixed poison rate (limiting the number of mislabeled images), our goal is to maximize both ASR and CA. However, it is a challenge in clean-image backdoor settings, as we can only modify the labels of images, leading to a discrete hard-label issue. Even advanced discrete optimization methods like GCG can only maximize ASR but struggle to maintain a high CA.

Our observations lead us to model this problem as a divergence maximization problem constrained by three factors: (a) Existence: The trigger pattern must be present within the training data, enabling backdoor injection via label manipulation alone. (b) Separability: The images with and without the trigger must be distinctly separable, allowing easier backdoor learning and reducing the required poison rate. (c) Irrelevancy: The trigger should not interfere with benign class features to prevent a significant CA drop, as feature overlap can disrupt class semantics. To satisfy these constraints, we introduce Conditional Information Maximizing GANs (C-InfoGAN). In C-InfoGAN, we introduce a discrete random variable $c$ following a Bernoulli distribution as the latent variable. The generator $G$, conditioned on $c$, generates two distinct series of images depending on whether $c$ is 0 or 1.

**(a) Existence.** A crucial property of clean-image backdoors is that the trigger pattern must exist within the clean image set. To satisfy this, we employ a standard GAN framework (Goodfellow et al., 2014). Training the discriminator $D$ ensures that all images generated by the generator $G$ follow the same distribution as real images. By conditioning $G$ on the latent variable $c$, we can generate images with ($c = 1$) or without ($c = 0$) the trigger pattern. Consequently, one of the two image series

generated, $P(\hat{x}|c=1)$, becomes a subset of the real image distribution. This series, $P(\hat{x}|c=1)$, can thus be safely used as the trigger function, guaranteeing its existence within the original image set.

**(b) Separability.** To ensure separability, we follow the concept of InfoGAN (Chen et al., 2016), which maximizes the mutual information between selected latent variables and the generated data to learn interpretable and disentangled representations. The recognition network $Q$ (originating from InfoGAN) is tasked with distinguishing between images generated with $c=1$ and $c=0$ as accurately as possible by introducing an information loss term $L_{\text{info}}$. $Q$ converges once it can easily determine which series an image belongs to, indicating strong separability.

**(c) Irrelevancy.** Another crucial attribute of backdoors is that the trigger should not interfere with the benign task. This indicates that the trigger pattern needs to be irrelevant to the patterns utilized for the benign task. To ensure this, we use the input image's ground-truth label $y$ as an auxiliary input to both the GAN generator $G$ and discriminator $D$, along with the condition variable $c$, ensuring $c$ is independent of $y$. Thus, when $c=1$ (triggered image), the generated image is unrelated to the input image class, minimizing the trigger's impact on the benign task.

**Objective Function.** In practice, our loss function combines the GAN loss from Wasserstein GAN (Arjovsky et al., 2017) and the mutual information loss from InfoGAN (Chen et al., 2016). The GAN loss is $L_{GAN} = \mathbb{E}_{\hat{x} \sim P_g}[D(\hat{x})] - \mathbb{E}_{x \sim P_X}[D(x)]$, where $P_X$ represents the distribution of real inputs and $P_g$ denotes the distribution generator's outputs, penalizing for the low consistency between these two distributions. The mutual information loss is $L_{info} = -\mathbb{E}_{c \sim P_c, x \sim P_X}[\log Q(c|G(x,c))]$, where $P_c$ is the Bernoulli distribution of $c$, represent the negative log-likelihood for predicting $c$ based on generated images $G(x,c)$. The overall loss function integrates these two components as $L = L_{GAN} + \lambda L_{info}$, where $\lambda$ is the trade-off hyperparameter.

**Theory Provement.** We also provide a theoretical analysis for our GCB attack in Appendix A. From the perspective of information theory, we show that minimizing information loss can maximize the JS-Divergence between $G(\cdot, c=0)$ and $G(\cdot, c=1)$ (Lemma A.1). In addition, GAN ensures that the generated images $G(x, \cdot)$ and the corresponding real images $x$ share the same distribution. Consequently, the JS-Divergence between the real images of the two series is maximized, allowing the $Q$ component to distinguish them effectively. Both parts mentioned above act as a special case of InfoGAN (Chen et al., 2016) and can be easily proved. Moreover, we prove that the total conditional entropy of the backdoor task: $H(Y'|X)$ (where $(X, Y')$ is the poisoned dataset), is minimized when the divergence between real images of the two series is maximized (Proposition A.2). Such minimizing indicates that our backdoor task is easily learned, thereby ensuring a high ASR.

### 4.3 ATTACK DEPLOYMENT

**Poisoning Stage.** We select a subset $X_1$ from the original training set $X$ and change their labels to the target label $y_t$. The key challenge is selecting which images to manipulate. We introduce a score function to assign poison scores to each clean image, where a higher score indicates greater suitability for label manipulation. The recognition network $Q$ from InfoGAN effectively serves as this score function. $Q$ is trained to recognize the value of $c$ in generated images $\hat{x}$. Since the GAN has converged, $x$ and $\hat{x}$ follow the same distribution, allowing $Q$ can recognize both generated $\hat{x}$ and real images $x$. After scoring all input images, we apply a top-$k$ quantile threshold to select the top-scoring images, where $k$ is the total number of poisoned samples needed. These selected images have their labels flipped and are then submitted to train the victim model.

**Inference Stage.** During the inference stage, to create a triggered image, we input any image $x$ into the generator $G$ conditioned on $c=1$, producing $G(x, c=1)$, which contains the trigger pattern. The triggered images exactly correspond to the selected images in the poisoning stage, thereby effectively activating the backdoor to mislead the victim model into predicting the target label $y_t$.

## 5 EVALUATION

### 5.1 EXPERIMENTAL SETUP

**Datasets and Models.** We evaluate our attacks using BackdoorBench (Wu et al., 2022) on five datasets: MNIST (LeCun et al., 1998), CIFAR-10/100 (Krizhevsky, 2009), GTSRB (Stallkamp et al.,

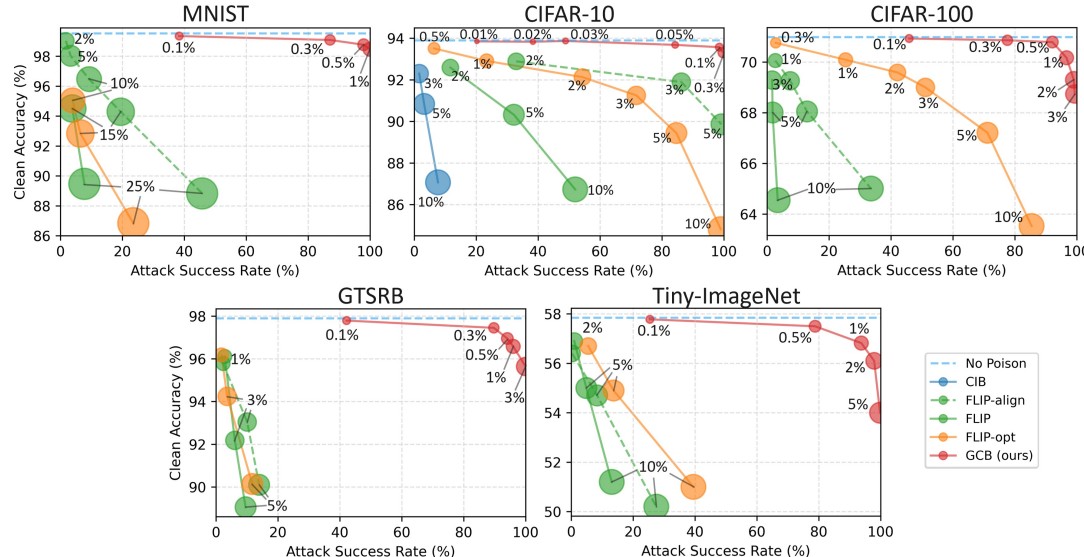

Figure 3: Performance for clean-image backdoor methods across various datasets. Poison rates are indicated by marker size and along with texts on each point. Our method, GCB, achieves success with less than 1% drop in clean accuracy to achieve attack success rate over 90% for all datasets.

2012), and Tiny-ImageNet (Russakovsky et al., 2015). We employ PreActResNet18 as the default victim model with a poison rate of 1%, if not specified. All results follow an all-to-one attack scenario. Detailed training settings for C-InfoGAN are provided in Appendix B.

**Baselines.** Our clean-image backdoor baselines include CIB (Chen et al., 2022a), FLIP (Jha et al., 2024), CIBA (Rong et al., 2024), and FLIP-opt. CIBA exhibits low ASR and does not release its code, so its performance is shown only in Appendix C.2. FLIP-opt combines FLIP and Narcissus (Zeng et al., 2023) for trigger optimization. Specifically, we first generate an optimized trigger using Narcissus, then determine the best label assignments for poisoning using FLIP. Additionally, we found that FLIP is highly sensitive to the victim model's architecture, relying on alignment between the victim and surrogate models used in attack preparation. A detailed analysis of this effect is in Appendix 8. To ensure a fair comparison, we report FLIP results under both aligned and unaligned conditions, labeled as FLIP-align and FLIP.

**Metrics.** We use two metrics in our experiments: *Clean Accuracy* (CA) and *Attack Success Rate* (ASR). CA measures the victim model's accuracy on clean test data, while ASR indicates the percentage of test instances with embedded triggers that are classified as the target class by the model.

## 5.2 ATTACK PERFORMANCE.

**ASR VS. CA.** We compare GCB with several clean-image backdoor baselines in Fig. 3. Our experiments demonstrate that GCB significantly outperforms all baselines across all datasets. With less than a 0.5% drop in CA, GCB achieves over 90% ASR on small datasets such as MNIST, CIFAR-10, and CIFAR-100. For more complex datasets like GTSRB and Tiny-ImageNet, GCB maintains over 90% ASR with a CA drop within 1%. In contrast, all tested baselines only succeed on simple datasets like CIFAR-10 and CIFAR-100, incurring CA drops exceeding 5%. Moreover, they fail on relatively complex datasets such as GTSRB and Tiny-ImageNet, and surprisingly even on the simple MNIST dataset. This failure on MNIST is likely because MNIST consists of grayscale, feature-poor images. Consequently, intuitively selected triggers (e.g., sinusoidal triggers) cannot be effectively constructed using clean image combinations.

**Convergence Speed.** Our key idea is to make the trigger easier for the victim model to learn by optimizing separability. An important question is how quickly the victim model can learn this trigger. Fig. 4 shows that our method converges to nearly 100% ASR in just 4 epochs, whereas the simplest backdoor attack, BadNets, requires 11 epochs to converge. This indicates that our backdoor task is even easier for neural networks than BadNets. Compared to peer clean-image backdoor methods, FLIP takes over 20 epochs to achieve a successful attack and remains unstable after 20 epochs.

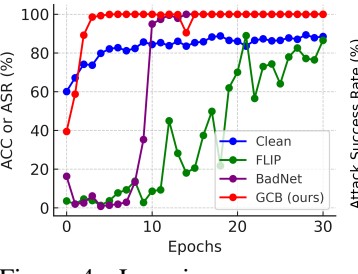
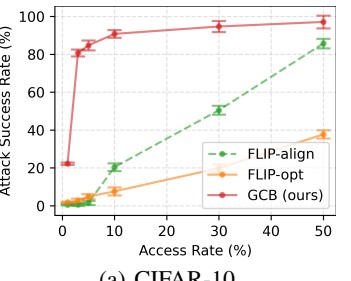
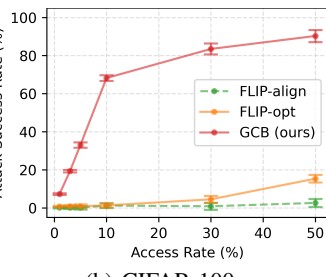

Figure 4: Learning curves on CIFAR-10. Our method converges even faster than BadNets.

(a) CIFAR-10

(b) CIFAR-100

Figure 5: Results with error bars under low access rates.

Table 2: Performance on Multi-Label Classification datasets. src denotes source class. GCB attack successfully with almost no drop in MAP.

| Method | Metric | VOC07 | VOC12 |
|--------|--------|-------|-------|
| CIB | ASR ↑ | 87.5±14.2 | 85.2±13.0 |
| | MAP ↑ | 91.8±1.1 | 91.3±1.4 |
| | MAP (src) ↑ | 74.8±3.1 | 72.6±4.9 |
| GCB | ASR ↑ | 67.5±7.2 | 70.1±8.5 |
| | MAP ↑ | 93.9±0.3 | 93.7±0.4 |
| | MAP (src) ↑ | 93.5±0.3 | 93.4±0.3 |

Table 3: Performance of GCB on other vision tasks. AE: Attack Mean Square Error. CE: Clean Mean Square Error.

| Task Dataset | Regression ColorCIFAR10 | | Segmentation VOC2012 | |
|--------------|------|------|------|------|
| Metrics | AE ↓ | CE ↓ | AE ↓ | CE ↓ |
| Clean | 0.2964 | 0.0128 | 1.207 | 0.211 |
| 1% Poison | 0.0290 | 0.0141 | 0.303 | 0.214 |
| 3% Poison | 0.0204 | 0.0156 | 0.277 | 0.217 |

**Impact of Model Architecture.** As introduced in the baseline settings, FLIP is highly sensitive to the victim model's architecture. In contrast, GCB exhibits high ASR across four distinct architectures: PreActResNet18, EfficientNet-B0, VGG-11, and ViT-B-16, as shown in Table 10. In our experiments, all four architectures achieve ASR exceeding 90% on every tested dataset, with an average ASR above 96%. This demonstrates that our method is architecture-agnostic.

**Generalized Threat Model.** Our threat model can be extended to weaker assumptions. We propose a generalized threat model where attackers can access only a small portion of the entire dataset and subsequently poison an even smaller subset of the accessed data. This extension broadens the clean-image backdoor threat to individual annotators with very limited dataset access. As shown in Fig. 5, when accessing only 10% of the training dataset, GCB achieves an ASR of 90.3% on CIFAR-10 and 68.2% on CIFAR-100. In comparison, the current SOTA baseline FLIP achieves only 20.4% and 1.3% ASR on CIFAR-10 and CIFAR-100, respectively, with the same data access.

**Other Vision Tasks.** Our method (GCB) is adaptable to various supervised vision tasks because C-InfoGAN is designed without specific assumptions about the target task. We simply adjust the label condition $y$ for different tasks—using one-hot encoding for classification and no embedding for regression—enabling seamless adaptation. For multi-label classification, we compared our approach with CIB (Chen et al., 2022a) using a 5% poison rate on the VOC07 and VOC12 datasets. As shown in Table 3, CIB achieves approximately 15% higher ASR but significantly underperforms in Mean Average Precision (MAP), dropping by about 2% overall and around 20% for the source class. This reduction in MAP compromises its stealthiness. Additionally, our method extends to Image Regression and Semantic Segmentation

Table 4: Ablation Study.

| poison rate | 1% | 0.5% | 0.1% |
|-------------|-----|------|------|
| **CIFAR-10** | | | |
| *(w/o $L_{GAN}$)* | 8.97 | 4.14 | 1.90 |
| *(w/o $L_{info}$)* | 42.9 | 11.4 | 2.87 |
| *(w/o LC)* | 98.9 | 93.1 | 85.3 |
| *Ours* | **100.0** | **100.0** | **98.5** |
| **CIFAR-100** | | | |
| *(w/o $L_{GAN}$)* | 3.41 | 1.80 | 0.45 |
| *(w/o $L_{info}$)* | 28.7 | 8.12 | 1.34 |
| *(w/o LC)* | 84.7 | 68.4 | 34.6 |
| *Ours* | **96.7** | **92.1** | **45.9** |

tasks, where existing clean-image backdoors are ineffective. As illustrated in Table 3, our attack succeeds in these tasks, demonstrated by a substantial decrease in Attack Mean Square Error (AE) compared to the clean dataset. Detailed task configurations are provided in Appendix C.4.

**Ablation Study.** We conducted ablation studies on three key components of our design: GAN loss (for Existence), information loss (for Separability), and label condition (for Irrelevancy). The results are presented in Table 4. (a) **GAN Loss.** We eliminate the discriminator $D$ from C-InfoGAN and apply an $l_\infty$-norm constraint to the generator. Experiments show that this approach completely loses

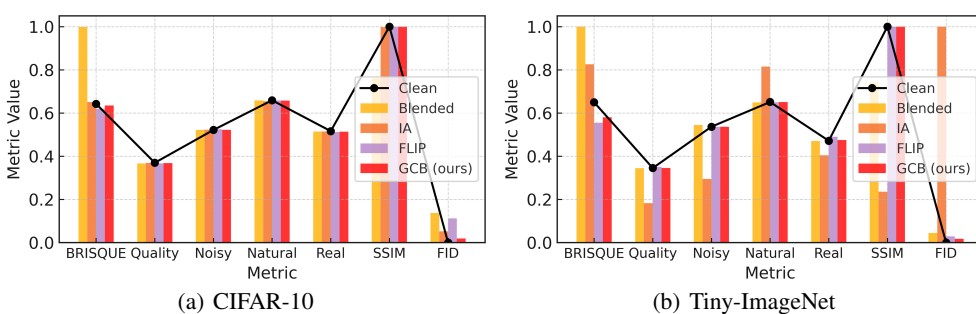

(a) CIFAR-10            (b) Tiny-ImageNet

Figure 6: Difference from clean images. Closeness to "Clean" values indicates stealthiness.

effectiveness because, without adversarial training, the trigger feature quickly overfits and becomes an adversarial attack on the recognition network $Q$, ceasing to function as an effective backdoor. (b) **Information Loss.** Removing the information loss transforms our network into a standard Pix2Pix GAN. To perform the attack, we intuitively select the darkest 1% of images in the dataset as poisoned images to construct the trigger feature, modeling the trigger-wrapping problem as a style-transfer scenario solvable by Pix2Pix GAN. Under this setup, GCB significantly degrades in performance, indicating that manually designed triggers are ineffective. (c) **Label Condition.** We remove $y$ as a prior condition from all components in C-InfoGAN. The results show only a slight decrease in ASR, likely because the UNet generator preserves the original appearance, diminishing the importance of the label condition.

## 5.3 Difference from Clean Images

To study how different trigger images from clean images are, we employ seven metrics introduced by BackdoorBench (Wu et al., 2024). Specifically, SSIM measures the structural similarity between each poisoned sample and its corresponding clean sample. FID assesses the distance between the distributions of poisoned samples and target-class clean samples. Additionally, BRISQUE, along with metrics for quality, noisiness, naturalness, and realism, evaluates the quality of individual samples. As shown in Fig. 6, clean-image backdoor attacks, such as FLIP and GCB, achieve better stealthiness (close to "Clean" line) compared to poison-image backdoors like Blended (Chen et al., 2017) and IA (Nguyen and Tran, 2020a). This is because clean-image backdoors utilize specific benign features to construct the backdoor, resulting in poisoned data that closely resemble benign images in quality and attributes. Thus reduce the effectiveness of image-quality-based detection methods including BRISQUE, Quality, Noisy, Natural, Real and SSIM. On the other hand, FID assesses distributions and can potentially detect differences between benign and triggered sample distributions caused by clean-image backdoors. While, our experiments demonstrate that GCB causes triggered images to closely resemble clean images in all tested metrics, including FID.

## 6 Defenses

### 6.1 Existing Defenses

We present the most representative defense methods in this section. Comprehensive evaluations of GCB against BackdoorBench defenses are provided in Appendix D.2.

**Neural Cleanse.** Neural Cleanse (Wang et al., 2019) uses anomaly scores to detect backdoors in DNN models. However, Fig. 7 shows that Neural Cleanse is hard to differentiate backdoor-attacked datasets and clean ones, because their scores are similar and below the 2.0 threshold. This is due to Neural Cleanse's focus on static adversarial patches, while our attack uses a dynamic, global trigger function, making trigger reconstruction difficult.

**STRIP.** STRIP (Gao et al., 2019) measures class prediction entropy through input perturbations. Fig. 8 shows a notable similarity in entropy distribution for clean and poisoned subsets. Since C-InfoGAN uses benign features of various intensities as triggers, it can yield similar STRIP behaviors for samples with or without trigger. Therefore, our GCB attack is resilient to STRIP defense.

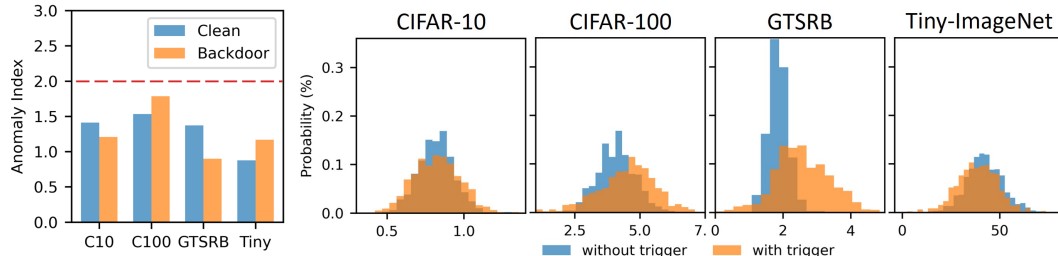

Figure 7: Neural Cleanse

Figure 8: STRIP normalized entropy distribution of GCB.

Table 5: Comparison of different attack methods against preprocessing-based defenses.

| Defense | No defense | | ShrinkPad | | Compression | | Color Shift | | DeepSweep | | Average |
|---|---|---|---|---|---|---|---|---|---|---|---|
| Attack | CA | ASR | CA | ASR | CA | ASR | CA | ASR | CA | ASR | Avg. ASR |
| BadNet | 93.2 | 73.8 | 83.8 | 60.5 | 39.9 | 3.3 | 81.9 | 53.1 | 85.3 | 1.9 | 29.7 |
| Blended | **93.8** | 94.1 | 84.2 | 85.2 | 42.6 | 2.6 | 86.3 | 86.9 | 70.9 | 65.5 | 60.0 |
| SIG | 93.7 | 80.4 | 84.0 | 85.7 | 43.9 | 70.1 | **86.8** | 74.3 | 84.5 | 41.6 | 67.9 |
| IA | 90.5 | 79.6 | 80.8 | 19.4 | 33.0 | 2.6 | 78.7 | 65.1 | **87.6** | 65.9 | 38.3 |
| SSBA | 93.4 | 99.7 | 83.5 | 1.5 | 37.4 | 18.5 | 86.2 | 94.2 | 71.8 | 81.2 | 48.9 |
| FLIP | 91.9 | 86.3 | **89.7** | 84.2 | **72.5** | 86.9 | 83.7 | 83.8 | 70.7 | 26.9 | 70.5 |
| Ours | 92.6 | **100.0** | 82.6 | **100.0** | 40.7 | **100.0** | 84.4 | **98.4** | 77.7 | **93.2** | **97.9** |

**Fine-Pruning.** Fine-Pruning (Liu et al., 2018) tries to mitigate backdoor behaviors by pruning high-activation neurons. As shown in Fig. 9, on CIFAR-10, the ASR remains unchanged regardless of pruning. In contrast, for CIFAR-100, ASR initially decreases but then rapidly increases as more neurons are pruned. Our backdoor attack leverages natural benign features, resulting in a robust and complex activation pattern that Fine-Pruning cannot detect. This suggests Fine-Pruning is ineffective against our backdoor attack.

**Grad-Cam.** We employ Grad-Cam (Gildenblat and contributors, 2021) to visualize the regions of an image that are most relevant to a model's prediction. Grad-Cam can also highlight potential trigger regions activated by different backdoor attacks. Results in Fig. 10 reveal that GCB's triggers are dispersed and centrally located, whereas the triggers of other attacks are localized and prominent. This indicates that GCB achieves global feature dominance, diverging from Grad-Cam's localized focus, and thus is more resilient to Grad-Cam-based detection.

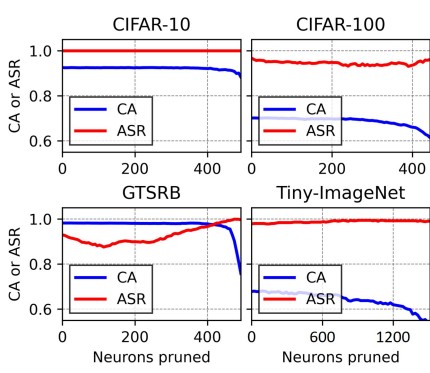

Figure 9: Fine-pruning.

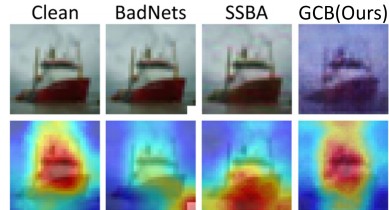

Figure 10: Grad-Cam.

### 6.2 TEST-TIME TRANSFORMATION DEFENSE

Although clean-image backdoors do not poison images during training, triggers are still used to activate the backdoor at test time. To evaluate test-time preprocessing defenses, we assess several prominent techniques in our experiments: 1.*ShrinkPad* (Li et al., 2020b): Pads testing images with zero-valued pixels after a 2-pixel shrinkage. 2. *Image Compression* (Xue et al., 2023): Applies JPEG compression to all testing images at 75% quality. 3. *Color Shift* (Jiang et al., 2023): Introduces a random color space shift between -0.1 and 0.1, specifically targeting color-based backdoors in datasets like CIFAR-10. 4. *DeepSweep* (Qiu et al., 2021): Uses 4 data augmentation methods in DeepSweep to fine-tune the victim model for 5 epochs and preprocesses the testing samples accordingly.

Table 5 presents the results of these preprocessing defenses on the CIFAR-10 dataset. Additive trigger-based attacks, such as BadNets, show reduced ASR when subjected to image transformations. Natural

trigger-based attacks like SIG and FLIP remain robust against most defenses but are compromised by image compression. In contrast, our GCB attack maintains nearly 100% ASR across all preprocessing defenses. This resilience is attributed to using a dominant semantic feature as the trigger, which is more robust than more fragile and intricate label features. Additional results on CIFAR-100 dataset are available in Appendix E.1.

## 6.3 Adaptive Defenses

**Noisy Training.** Clean-image backdoors embed triggers by poisoning only labels. Consequently, training techniques that are robust to label noise might diminish the effectiveness of these faulty labels. We evaluated three noisy training methods: Self-Paced Learning (SPL) (Kumar et al., 2010), Perturbation Robust Learning (PRL) (Wong and Kolter, 2020), and Bootstrap (Reed et al., 2014). Results were recorded at the final epoch and at the best epoch

Table 6: Noisy training mitigation. Data is recorded in CA/ASR format. C10: CIFAR-10. C100: CIFAR-100.

|            | SPL       | PRL       | Bootstrap |
|------------|-----------|-----------|-----------|
| C10-final  | 91.9/100  | 89.7/100  | 88.4/100  |
| C10-best   | 36.0/6.1  | 59.3/65.5 | 26.0/0.0  |
| C100-final | 67.2/78.2 | 66.8/87.6 | 57.6/93.9 |
| C100-best  | 36.5/4.9  | 51.3/21.6 | 14.0/0.8  |

where the ratio $\frac{CA}{ASR}$ was highest. As shown in Table 6, none of these methods effectively defend against our attack. This is likely because GCB's incorrect labels constitute misleading knowledge rather than random noise, which contradicts the basic assumption of noisy training.

**Advanced Label Cleaning.** Advanced label cleaning techniques can automatically identify potentially mislabeled data in a suspicious dataset. These methods typically involve training a model and flagging images with low confidence scores as having label issues. We employ CleanLab, a widely used tool with strong community support (9.4k GitHub stars), to evaluate its effectiveness in defending against GCB. Using CleanLab, we analyze the confidence score distribution, as shown in Fig. 11. Noisy labels are easily separated because they exhibit very low confidence scores, as their label issues are random and the DNN cannot establish a reliable mapping between images and their corresponding labels. In contrast, GCB selects specific images to bind to target labels, creating a strong connection and resulting in higher confidence scores than benign images, as illustrated in Fig. 11(b). Therefore, advanced label cleaning techniques are ineffective against GCB.

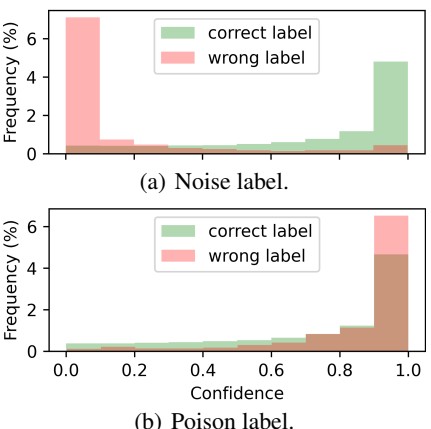

Figure 11: Confidence distributions for two different kinds of label issues.

**Relabeling Mitigation.** Relabeling is the most direct mitigation strategy against GCB. Since only labels are altered in the poisoned dataset, relabeling a portion can restore its integrity, thereby increasing CA and reducing ASR. Fig. 12 shows relabeling results on CIFAR-10 dataset, with each data point marked by upper and lower bounds of five independent trials. Tests at poison rates of 0.1%, 0.5%, and 1% indicate that higher poison rates diminish relabeling's effectiveness. To ensure model security and keep ASR below 20%, a relabeling rate above 95% is necessary, implying that nearly the entire dataset must be relabeled.

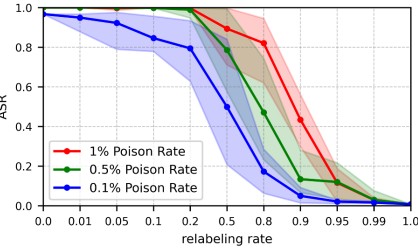

Figure 12: Relabeling Mitigation.

## 7 Conclusion

We introduced Generative Adversarial Clean-Image Backdoors (GCB), a stealthy and adaptive backdoor attack that uses C-InfoGAN to optimize trigger patterns embedded within training images. Experiments across 5 datasets, 5 models, and 4 tasks showed high attack success rates with minimal drop in clean accuracy and low poison rates. GCB resists existing defenses, highlighting the need for more robust protections.

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

# A  MATHEMATICAL ANALYSIS

In this section, our goal is to demonstrate *why optimizing C-InfoGAN leads to the optimization of our clean-image backdoor task*. It is worth noting that this proof is primarily based on the standard GAN loss; however, in practice, we utilize the Wasserstein GAN loss for more stable training.

## A.1  ANALYSIS ON C-INFOGAN

**Lemma 1** C-InfoGAN's loss function is equivalent to maximize **weighted Jensen–Shannon divergence** between $p(\hat{x_0})$ and $p(\hat{x_1})$, where $\hat{x_0} = G(x, c = 0)$ and $\hat{x_1} = G(x, c = 1)$, $G(x, c) = p(\hat{x}) = p(\hat{x_0}) \cup p(\hat{x_1})$, and $x \in X$ is original image data.

**Proof** C-InfoGAN derives its loss function directly from InfoGAN, where the mutual information loss $I(c, G(x, c))$ is maximized. Mutual information loss is defined by:

$$I(c, G(x, c)) = \sum_c \sum_{\hat{x}} p(c, \hat{x}) \log \frac{p(c|\hat{x})}{p(c)} = \sum_{\hat{x}} \sum_c p(c) p(\hat{x}|c) \log \frac{p(\hat{x}|c)}{p(\hat{x})}$$

Assume that $p(c)$ is a known Bernoulli distribution with $p(c = 1) = p_r$ and $p(c = 0) = 1 - p_r$ respectively. $I(c, G(x, c))$ can be expanded in two terms.

$$I(c, G(x, c)) = \sum_{\hat{x}} \left[ p(c = 0) p(\hat{x}|c = 0) \log \frac{p(\hat{x}|c = 0)}{p(\hat{x})} + p(c = 1) p(\hat{x}|c = 1) \log \frac{p(\hat{x}|c = 1)}{p(\hat{x})} \right]$$

$$= (1 - p_r) \operatorname{KL} \left( p(\hat{x}|c = 0) \parallel p(\hat{x}) \right) + p_r \operatorname{KL} \left( p(\hat{x}|c = 1) \parallel p(\hat{x}) \right)$$

$$= (1 - p_r) \operatorname{KL} \left( p(G(x, c = 0)) \parallel p(\hat{x}) \right) + p_r \operatorname{KL} \left( p(G(x, c = 1)) \parallel p(\hat{x}) \right)$$

$$= (1 - p_r) \operatorname{KL} \left( p(\hat{x_0}) \parallel p(\hat{x}) \right) + p_r \operatorname{KL} \left( p(\hat{x_1}) \parallel p(\hat{x}) \right)$$

Where KL denotes Kullback-Leibler divergence. Noticed that $p(\hat{x}) = p(\hat{x_0})(1 - p_r) + p(\hat{x_1}) p_r$, so the above equation aptly fit the format of **weighted Jensen–Shannon divergence**.

$$I(c, G(x, c)) = \operatorname{JSD}_{1 - p_r, p_r} \left( p(\hat{x_0}) \parallel p(\hat{x_1}) \right)$$

So, maximizing mutual information term $I(c, G(x, c))$ is equivalent to maximizing JS divergence between two series of generated images $p(\hat{x_0})$ and $p(\hat{x_1})$ with weight exactly equals to poison rate $p_r$.

**Lemma 2** When C-InfoGAN is sufficiently converged, both $\operatorname{JSD}(p(\hat{x_1}) \parallel p(x_1))$ and $\operatorname{JSD}(p(\hat{x_0}) \parallel p(x_0))$ will be minimized to the limit of zero, where $\hat{x_0} = G(x, c = 0)$ and $\hat{x_1} = G(x, c = 1)$ denote two series of generated data. $p(x_0) = \{x \in X | s(x) < 0\}$ and $p(x_1) = \{x \in X | s(x) \geq 0\}$ denote real data partitioned by $s(x)$.

**Proof** We first take GAN term into consideration

$$\min_G \max_D L_{\operatorname{GAN}}(G, D) = \mathbb{E}_{x \sim p(x)}[\log D(x)] + \mathbb{E}_{\hat{x} \sim p(\hat{x})}[\log(1 - D(\hat{x}))]$$

According to original GAN's preposition, when the discriminator is optimal, the equation can be re-expressed using the logistic sigmoid function $\sigma$ as follows:

$$D^*(x) = \frac{p(x)}{p(x) + p(\hat{x})}$$

Substitute $D^*(x)$ into the GAN objective function $L_{GAN}(D, G)$ gives:

$$L_{\operatorname{GAN}}(G, D^*) = \sum_x p(x) \log \left( \frac{p(x)}{p(x) + p(\hat{x})} \right) + p(\hat{x}) \log \left( \frac{p(\hat{x})}{p(x) + p(\hat{x})} \right)$$

This expression can then be recognized as the standard Jensen-Shannon divergence (JSD):

$$\operatorname{JSD}(p(x) \| p(\hat{x})) = \frac{1}{2} \operatorname{KL}(p(x) \| M) + \frac{1}{2} \operatorname{KL}(p(\hat{x}) \| M)$$

$$M = \frac{1}{2}(p(x) + p(\hat{x}))$$

So, $\min_G L_{\text{GAN}}(G, D^*)$ is just equivalent to minimize $\text{JSD}(p(x)||p(\hat{x}))$. When sufficient convergence occurs, the generated data distribution is infinitely close to the real data distribution, which means $\text{JSD}(p(x)||p(\hat{x})) \to 0$. Moreover, considering the mutual information term and its variational lower bound $L_I(G, Q)$

$$I(c; G(x, c)) \geq L_I(G, Q) = \mathbb{E}_{\hat{x} \sim G(x,c)}[\mathbb{E}_{c' \sim p(c|\hat{x})}[\log Q(c'|\hat{x})]] + H(c)$$

is exactly maximizing log likelihood to accurately estimate $P(c|\hat{x})$ using $Q(c|\hat{x})$. So when Q is optimal,

$$Q^*(c = 1|\hat{x}) = P(c = 1|\hat{x}) = P(c = 1|G(z, c)) = \left\{ \begin{array}{ll} 1, & \text{if } \hat{x} \sim p(\hat{x_1}) \\ 0, & \text{if } \hat{x} \sim p(\hat{x_0}) \end{array} \right.$$

$$Q^*(c = 0|\hat{x}) = \left\{ \begin{array}{ll} 0, & \text{if } \hat{x} \sim p(\hat{x_1}) \\ 1, & \text{if } \hat{x} \sim p(\hat{x_0}) \end{array} \right.$$

We further define $s(x)$ as subtracting the above two term and we have:

$$s(x) = Q^*(c = 1|x) - Q^*(c = 0|x) = \left\{ \begin{array}{ll} 1, & \text{if } x \sim p(\hat{x_1}) \\ -1, & \text{if } x \sim p(\hat{x_0}) \end{array} \right.$$

Then, both the $p(\hat{x_1})$ and $p(x_1)$ can be regarded as a sub-distribution of $p(\hat{x})$ and $p(x)$ that partitioned by a function $s(x)$. For $p(\hat{x_1})$,

$$\hat{x_1} = G(x, c = 1) = \{x \sim p(\hat{x})|s(x) \geq 0\}$$

For $p(x_1)$, it is defined as $p(x_1) = \{x \sim p(x)|s(x) \geq 0\}$. Finally, we can multiplying a step function $\mathbb{1}(s(x) \geq 0)$ for all possible $x$ in $p(\hat{x})$ and $p(x)$ and get:

$$\text{JSD}(p(x_1)||p(\hat{x_1})) = \frac{1}{2}\lambda \sum_x \mathbb{1}(s(x) \geq 0) \left( p(x) \log \left( \frac{2p(x)}{p(x) + p(\hat{x})} \right) + p(\hat{x}) \log \left( \frac{2p(\hat{x})}{p(x) + p(\hat{x})} \right) \right)$$

$$\leq \frac{1}{2}\lambda \sum_x p(x) \log \left( \frac{2p(x)}{p(x) + p(\hat{x})} \right) + p(\hat{x}) \log \left( \frac{2p(\hat{x})}{p(x) + p(\hat{x})} \right)$$

$$= \lambda \cdot \text{JSD}(p(x)||p(\hat{x}))$$

where $\lambda > 1$ denotes a normalization factor so that $\lambda \cdot \mathbb{1}(s(x) \geq 0)p(x)$ can still be a distribution. The result shows that $\lambda \cdot \text{JSD}(p(x)||p(\hat{x}))$ is exactly the upper bound of $\text{JSD}(p(x_1)||p(\hat{x_1}))$. This bound is tight because $\text{JSD}(p(x)||p(\hat{x})) \to 0$ during optimization and $\text{JSD}(p(x_1)||p(\hat{x_1})) \geq 0$. Thus minimizing $\text{JSD}(p(x)||p(\hat{x}))$ to a limit of zero would also minimize $\text{JSD}(p(x_1)||p(\hat{x_1}))$ to a limit of zero. Similar process can be applied to $\text{JSD}(p(x_0)||p(\hat{x_0}))$ and get the same result.

**Proposition 1** When C-InfoGAN gets well-trained and converged. Then, the following hold: (a) $\text{JSD}(p(x_1) \parallel p(x_0))$ is maximized. (b) $\text{JSD}(p(\hat{x_1}) \parallel p(x_1))$ is minimized to a limit of zero. where $\hat{x_0} = G(x, c = 0)$ and $\hat{x_1} = G(x, c = 1)$ denote two series of generated data. $p(x_0) = \{x \in X|s(x) < 0\}$ and $p(x_1) = \{x \in X|s(x) \geq 0\}$ denote real data partitioned by $s(x)$.

**Proof** (b) has already been proved in Lemma 2. (a) can be easily proved since the triangle inequality holds for the square root of JSD (Osán et al., 2018). Apply the triangle inequality to square root of $\text{JSD}(p(x_1) \parallel p(x_0))$ and we get:

$$\sqrt{\text{JSD}(p(x_1) \parallel p(x_0))} \geq \sqrt{\text{JSD}(p(\hat{x_1}) \parallel p(\hat{x_0}))} - \sqrt{\text{JSD}(p(x_0) \parallel p(\hat{x_0}))} - \sqrt{\text{JSD}(p(x_1) \parallel p(\hat{x_1}))}$$

According to lemma 1, $\text{JSD}(p(\hat{x_1}) \parallel p(\hat{x_0}))$ is maximized. According to lemma 2, $\text{JSD}(p(x_0) \parallel p(\hat{x_0}))$ and $\text{JSD}(p(x_1) \parallel p(\hat{x_1}))$ is minimized to a limit of zero. The bound is tight because as $\text{JSD}(p(x_0) \parallel p(\hat{x_0})) \to 0$ and $\text{JSD}(p(x_1) \parallel p(\hat{x_1})) \to 0$, it follows that $\text{JSD}(p(x_1) \parallel p(x_0)) \to \text{JSD}(p(\hat{x_1}) \parallel p(\hat{x_0}))$. Therefore, $\text{JSD}(p(x_1) \parallel p(x_0))$ is maximized.

**Corollary 1** When C-InfoGAN's generator $G$, discriminator $D$, and mutual information estimator $Q$ are trained conditional on ground truth label $y$, and it get well-trained and converged. Then, the following hold: (a) $\text{JSD}(p(x_1|y) \parallel p(x_0|y))$ is maximized. (b) $\text{JSD}(p(\hat{x_1}|y) \parallel p(x_1|y))$ is minimized. This is because all $G$, $D$, and $Q$ will be given $y$ as a part of input that irrelevant to $c$, so they all model the conditional distribution $p(x|y)$ instead of $p(x)$. As a result, all the conclusions in Proposition 1 can be adapted to the conditional distribution version under this setting.

## A.2 Analysis on Clean-Image Backdoor

**Proposition 2** In clean-image backdoor, if 1) poison rate given, and 2) poison strategy is flipping all labels of selected images to target class; then maximize Jensen–Shannon divergence between selected/unselected images $\max \mathrm{JSD}(p(x_1|y) \parallel p(x_0|y))$ can result in minimize conditional entropy $\min H(Y'|X)$. where $p(x_1)$ denotes the distribution of inputs selected for label modification. $p(x_0)$ denotes the unselected input distribution. $Y'$ denotes poisoned label distribution.

**Proof** Enhancing performance of Image Classification problem can be regarded as $\min H(Y'|X)$, which means maximizing information that can be inferred from input $X$ to poisoned label $Y'$. It can be transformed into label conditional entropy:

$$H(Y'|X) = -H(X) + H(X|Y') + H(Y')$$

where in Clean-Image Backdoor, $H(X)$ is always a constant, and $H(Y')$ reaches its lower bound in our strategy that flips all labels of selected images toward the target class. As a result, the problem becomes $\min H(X|Y')$, meaning maximizing information that can be inferred from label $Y'$ to input $X$.

For further convenience, we denote a split variable $c$ as follows:

$$c(x) = \begin{cases} 1 & \text{if } x \sim p(x_1) \\ 0 & \text{if } x \sim p(x_0) \end{cases}$$

where we have $p(x) = p(c=0)p(x_0) + p(c=1)p(x_1) = p_r p(x_1) + (1-p_r)p(x_0)$ based on this definition. Moreover, based on assumption 2) in this section, the relationship between poisoned label $y'$ and true label $y$ can be given as follows:

$$y' = \begin{cases} y_t & \text{if } c = 1 \\ y & \text{if } c = 0 \end{cases}$$

Here $y_t$ denotes target label, and this equation means if $c = 1$ (images in $X_1$), the label will be directly flipped to target class, otherwise it will follows the original label offered by the dataset. Therefore, $y'$ can be fully determined given $c$ and $y$, which means $H(Y'|C,Y) = 0$. We further conclude that $H(C,Y,Y') = H(C,Y)$.

Using the above condition, the conditional entropy of original problem $H(X|Y')$ can be expanded into this form:

$$\begin{aligned} H(X|Y') &= H(X|C,Y) + H(X,Y') - H(Y') - H(X,Y,C) + H(C,Y) \\ &= H(X|C,Y) + H(X,Y') - H(X,Y,Y',C) - H(Y') + H(C,Y,Y') \\ &= H(X|C,Y) - H(C,Y|X,Y') + H(C,Y|Y') \\ &\leq H(X|C,Y) + H(C,Y|Y') \end{aligned}$$

where $H(C,Y|Y')$ is a constant information loss term from $C,Y$ to $Y'$. Since $c$ is independent to $y$, $H(X|C,Y)$ can be directly transformed to Jensen-Shannon Divergence by definition:

$$\begin{aligned} H(X|C,Y) &= -\sum_x \sum_{c=0}^{1} p(c) \sum_{y=1}^{N} p(y)p(x|c,y) \log p(x|c,y) \\ &= -\sum_{x_1} p_r \sum_{y=1}^{N} p(y)p(x_1|y) \log p(x_1|y) - \sum_{x_0}(1-p_r)\sum_{y=1}^{N} p(y)p(x_0|y) \log p(x_0|y) \\ &= H(X|Y) - \mathrm{JSD}_{p_r,1-p_r}(p(x_1|y) \parallel p(x_0|y)) \end{aligned}$$

where $H(X|Y)$ is a constant in Clean-Image Backdoor. As a result, $\max \mathrm{JSD}(p(x_1|y) \parallel p(x_0|y))$ will minimize the upper bound of $H(Y'|X)$. Moreover, this bound is tight once the JSD is optimized. $H(C|X) \to 0$ when JSD is optimized, at this time $C$ can be nearly fully determined given $X$. Based on this, we induce that $H(C,Y|X,Y') \to 0$, which proved the tightness of this bound when optimal.

The major goal of a clean-image backdoor is to maximize the Attack Success Rate (ASR) of the victim model on triggered test set, denoted as follows:

$$\max_{X_1,T} \mathbb{E}_{x \sim p(x)} \mathbb{1}[f^*(T(x)) = y_t]$$

$$f^*(x) = \operatorname*{argmin}_f [\sum_{(x_0, y_0) \in (X_0, Y_0)} L(f(x_0), y_0) + \sum_{x_1 \in X_1} L(f(x_1), y_t)]$$

where $X_1$ and $X_0$ denote partitioned poison dataset and benign dataset respectively, $y_t$ denotes target class, $T(\cdot)$ denotes trigger function, $L$ denotes loss function for victim model $f(\cdot)$. Let us try to solve this problem using C-InfoGAN. Let $T(x) = G(x, c = 1)$, $X_1 = \{x \in X | s(x) \geq 0\}$, $X_0 = \{x \in X | s(x) < 0\}$ We can consider this optimization problem in two steps:

**1. optimize $X_1$:** As discussed in **Proposition 2**, the problem of optimizing classification performance of Clean-Image Backdoor in image classification $\min H(Y'|X)$ can be transformed into divergence format $\max \mathrm{JSD}(p(x_1|y) \parallel p(x_0|y))$. This JS divergence can be exactly optimized by InfoGAN according to **Corollary 1**. As a result, we proved GCB to be effectively learned by the victim model from the perspective of Information Theory.

**2. optimize $T$:** The main goal of optimizing $T$ is to make $T(x)$ similar to $X_1$ to effectively activate the implanted backdoor. Meanwhile, $T(x)$ should preserve original semantic class information $y$, so the optimization problem could be rewrite as:

$$\min_T \mathrm{JSD}(p(T(x)|y) \parallel p(x_1|y))$$

Here according to our definition of $T(\cdot)$, $p(T(x)|y)$ can be computed as:

$$p(T(x)|y) = p(G(x, c = 1)|y) = p(\hat{x_1}|y)$$

Then according to Corollary 1. $\mathrm{JSD}(p(\hat{x_1}|y) \parallel p(x_1|y))$ would be minimized if C-InfoGAN converges. As a result, $T(\cdot)$ also reaches its optimality.

## B    HYPERPARAMETER SETTINGS

### B.1    C-INFOGAN SETTINGS

In our model, the ground-truth label $y$ is combined with the poison condition $c$, and integrated into the image feature map through cross-attention mechanisms (Vaswani et al., 2017) at each convolutional layer in the UNet structure. For all experiments, we apply batch normalization after most layers and set the random seed to 42 to ensure reproducible results. The temperature of the Gumbel Softmax (Jang et al., 2016) is set at 0.5. The batch size for all experiments is 256, with a weight decay of 1e-5. We use the Adam optimizer with betas of 0.5 and 0.999 for training for 100 epochs on each dataset. Both the generator and discriminator steps are set to 1. The gradient penalty weight for the Wasserstein GAN is consistently set at 10, adhering to the original setting in the Wasserstein GAN (Arjovsky et al., 2017). Following InfoGAN (Chen et al., 2016), we identified that certain parameters, such as the learning rate and information loss weight, are crucial for convergence. The hyperparameters for the learning rate and information loss weight for different structures are presented in Table 7. It is important to note that the provided hyperparameters are not the only set that can achieve convergence, but they demonstrate how to produce the results in this paper. All models are trained on Nvidia A100 for no more than 2 hours to get converged.

Table 7: Important hyperparameters setting in our experiment.

| Dataset | lr G | lr D | $\lambda$ |
|---|---|---|---|
| MNIST | 5e-4 | 1e-4 | 0.5 |
| CIFAR-10 | 4e-5 | 4e-5 | 0.25 |
| CIFAR-100 | 4e-4 | 2e-4 | 0.25 |
| GTSRB | 4e-5 | 4e-5 | 0.25 |
| ColorCIFAR10 | 4e-5 | 4e-5 | 0.25 |
| CelebA | 4e-4 | 4e-4 | 0.1 |
| VOC2012 | 4e-4 | 4e-4 | 0.1 |

### B.2    VICTIM MODEL SETTING

Unless specified, we use PreActResNet18 as default victim model, 0.01 as default poison rate. For training victim model, SGD with momentum of 0.9 is used under batch size of 128 and weight

decay of 0.0005. A cosine learning rate scheduler with an initial learning rate of 0.01 is also used for stable convergence. For CIFAR-10 and CIFAR-100, we use 100 epoch as default setting. For simpler datasets like MNIST or GTSRB, we use 20 and 50 as default epochs for quicker testing. All experiments on GCB are carried out in an all-to-one fashion.

# C  ADDITIONAL EVALUATIONS

## C.1  FLIP EVALUATION

**FLIP Experiment Setup.** We carry out our experiment based on the official code of FLIP and BackdoorBench [3]. In FLIP, the source class is set to all classes and the target class to class 0, aiming for an all-to-one attack. Poisoned labels for all samples are generated using FLIP and then sent to BackdoorBench for a fair comparison.

**Weakness of FLIP.** The major weakness of FLIP is that it is extremely sensitive to different architectures of victim models, as shown in Table 8. FLIP and our proposed method, GCB, were tested on different victim model architectures using CIFAR10 dataset. FLIP uses 3% poison rate and GCB uses 1% poison rate. Since FLIP's expert model defaults to PreActResNet34, it performs well on similar architectures like PreActResNet18. However, FLIP fails in poisoning for all other architectures, making it impractical in real scenarios because adversaries are unlikely to anticipate the victim model's structure.

Table 8: FLIP and our attack's performance on different architectures. FLIP only works well when victim model architecture is aligned with surrogate model architecture (e.g., PreActResNet18 in this case). On the contrary, our model works well on all architectures.

|  | Metrics | EfficientNet-B0 | PreActResNet18 | ResNet18 | VGG19 | Vit-B-16 |
|---|---|---|---|---|---|---|
| FLIP (3%) | ACC | 74.9% | 91.9% | 83.4% | 88.6% | 95.2% |
|  | ASR | 3.8% | 86.3% | 4.9% | 5.6% | 3.4% |
| GCB (1%) | ACC | 73.0% | 92.6% | 84.3% | 89.5% | 94.5% |
|  | ASR | 99.93% | 100.0% | 100.0% | 100.0% | 100.0% |

**Ablation Study of FLIP.** Note that all experiments in this paper are carried out in BackdoorBench under an all-to-one manner, so the result is slightly different from FLIP's official result (Appendix C.2.1 in their paper), which is tested under a one-to-one setting. We carried out an ablation study on attack mode (one-to-one / all-to-one) and victim model (r32p / VGG19). We use the official code from FLIP to implement all the experiments in this section. For one-to-one setting, the source class is set to 9 and target class of 4, while for all-to-one setting, the source class is set to all except 4 and target class of 4. We use an initial learning rate of 0.01 and 0.1 for VGG19 and r32p respectively. Note that VGG19 is treated as a large image model in FLIP's code so that input image is first resized to 224 and then passed to VGG19, while we directly pass it to VGG19 with size 32 for faster training. In FLIP's default setting, r32p is used for training expert models, so if the victim model also uses this or some similar architectures, the performance will be largely enhanced. As shown in Table 9, experiments with victim model r32p are much better than that of VGG19. The result becomes even worse when the attack mode is set to all-to-one attack, which means larger difficulty will be in this task. Here the ASR of (1), (3), (4) are very close to that of the original paper of FLIP. To conclude, the major reason for the inconsistency between our result and FLIP's official result is that the main experiments of our paper are carried out on the most difficult scenario (all-to-one and unaligned victim model), which is not studied in their official paper's results.

## C.2  ASR VS POISON RATES

We provide the Attack Success Rate versus poison rate as an additional result, particularly for positioning the CIBA attack on CIFAR-10 (see Fig. 13(b)). Since they do not release their code, we can only replicate their results here rather than directly compare them in the same benchmark as other attack methods in Fig. 3.

Table 9: Ablation Study of FLIP. (3) is used as the baseline in their original paper. (2) (4) are used as FLIP baseline (FLIP & FLIP-align respectively) in our paper. FLIP does not provide their model under scenario (2).

| ID | Attack Mode | Victim Model | Performance (ours) | | Performance (original paper) | |
|---|---|---|---|---|---|---|
| | | | CA | ASR | CA | ASR |
| (1) | one-to-one | vgg19 | 87.92% | 52.2% | 90.63% | 63.0% |
| (2) | all-to-one | vgg19 | 88.17% | 18.3% | N/A | N/A |
| (3) | one-to-one | r32p | 89.23% | 99.9% | 89.87% | 99.8% |
| (4) | all-to-one | r32p | 89.57% | 91.6% | 90.14% | 95.6% |

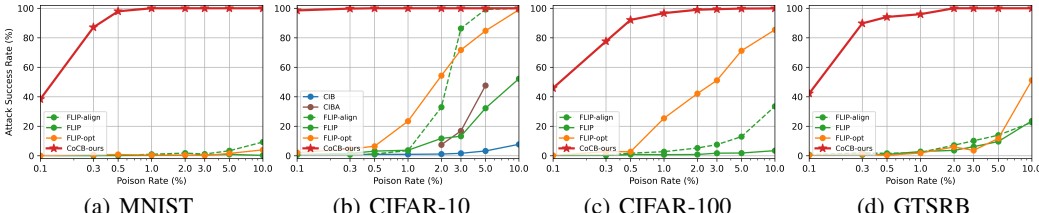

| (a) MNIST | (b) CIFAR-10 | (c) CIFAR-100 | (d) GTSRB |

Figure 13: ASR - poison rates for different attack methods

## C.3 DIFFERENT ARCHITECTURES

We evaluated our attack's robustness across various target model structures, including PreActResNet18(He et al., 2016), EfficientNet B0(Tan and Le, 2019), VGG11(Simonyan and Zisserman, 2015), and ViT-B-16(Dosovitskiy et al., 2020), chosen for their unique efficiencies, accuracies, and scalability. As Table 10 shows, our attack consistently achieves high ASR across these different architectures, implying model-agnostic traits. PreActResNet18, maintaining good CA while reaching the lowest ASR, is chosen as the basic architecture for all other experiments.

Table 10: Clean Accuracy (CA) and Attack Success Rate (ASR) of different architectures of poison rate 1%. Our model shows high ASR across all tested datasets and models.

| Architecture | PreActResNet18 | | VGG11 | | EfficientNet B0 | | ViT-B-16 | |
|---|---|---|---|---|---|---|---|---|
| Dataset | CA | ASR | CA | ASR | CA | ASR | CA | ASR |
| MNIST | 98.50 | 100.0 | 98.39 | 100.0 | 98.64 | 100.0 | 98.56 | 100.0 |
| CIFAR10 | 92.55 | 100.0 | 88.31 | 100.0 | 72.95 | 99.93 | 94.48 | 100.0 |
| CIFAR100 | 70.14 | 96.70 | 58.56 | 92.97 | 52.86 | 93.76 | 84.12 | 95.26 |
| GTSRB | 95.86 | 95.99 | 93.74 | 96.21 | 85.04 | 91.19 | 98.01 | 95.17 |
| Average | 89.26 | 98.17 | 84.75 | 97.30 | 77.37 | 96.22 | 93.79 | 97.60 |

## C.4 UNIVERSAL COMPUTER VISION TASK DETAILS

**CIB Details.** We carried out our multi-label experiment based on the official code of CIB (Chen et al., 2022a). We find that CIB can be highly sensitive to various source classes. To provide a more statistically significant result, we systematically tested each possible source class within the training dataset, calculating both the mean and standard deviation. For CIB one-to-one setting, we considered all label combinations with proportion of 5±1% as potential source classes.

**Dataset**. *Image Regression*: We introduce ColorCIFAR-10, derived from CIFAR-10 (Krizhevsky, 2009), with labels representing continuous features: hue, saturation, and illumination. Cyclic encoding is used for hue, resulting in four labels (sin hue, cos hue, saturation, illumination), each scaled to [-1, 1] with added Gaussian noise ($\mathcal{N}(0, 0.1^2)$ and clipped to [-1, 1].*Semantic Segmentation*: VOC2012 (Everingham et al., 2015) is used with a focus on samples with semantic segmentation annotations, totaling 2,330 training and 583 testing images.*Multi-label Binary Classification*: CelebA (Liu

et al., 2015) is utilized with five balanced and independent binary labels: Attractive, Mouth Slightly Open, High Cheekbones, Smiling, Wavy Hair.

**Architecture**. In training InfoGAN, cross-attention is employed for class feature encoding in all tasks. For Image Regression and Multi-label Binary Classification, labels are directly fed to cross-attention without preprocessing. For Semantic Segmentation, label images are added to the UNet image channels, bypassing cross-attention. Victim models are trained using PreActResNet18 for Image Regression and Multi-label Binary Classification, and UNet for Semantic Segmentation. All models undergo 100 epochs of training with SGD, an initial learning rate of 0.01, weight decay of 0.0005, and a standard cosine scheduler.

## D    POISON SAMPLES DISPLAY

### D.1    POISON SAMPLES IN IMAGE CLASSIFICATION

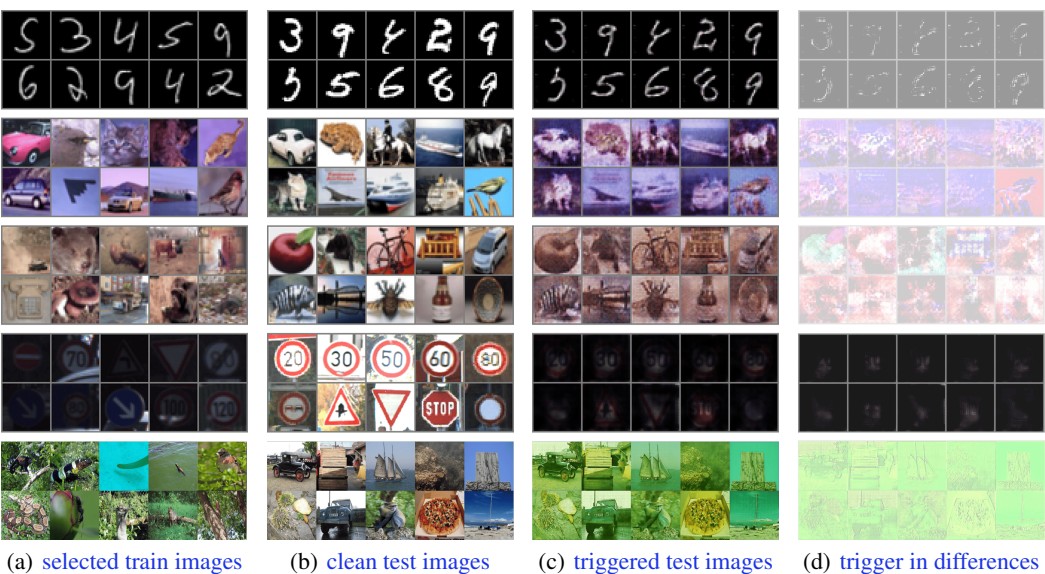

(a) selected train images    (b) clean test images    (c) triggered test images    (d) trigger in differences

Figure 14: Sample images from MNIST, CIFAR-10, CIFAR-100, GTSRB, and ImageNet-1K. The selected training images are unmodified, representing a subset of clean images.

As shown in Fig. 14, in our experiments, C-InfoGAN predominantly identified color as the trigger feature for all datasets except MNIST for its irrelevance to class information. Occasionally, this color trigger was combined with positional or global contrast features. For the MNIST dataset, where images are grayscale and normalized, the model adapted by learning more semantic features, such as the weight or thickness of digits, as triggers. Thus, C-InfoGAN is effective in identifying dominant semantic features unrelated to class labels as triggers.

### D.2    POISON SAMPLES IN OTHER TASKS

Analyzing the triggered features generated for each dataset reveals interesting distinctions. As shown in Fig. 15, VOC2012 retains color features similar to image classification tasks. In contrast, CelebA, where color might be label-relevant, learns background color as the triggered feature. Most notably, ColorCIFAR-10 selects image borders as the trigger, attributed to the prevalence of bordered images in the dataset. This suggests that InfoGAN can be directed to specific features by incorporating relevant priors, thereby avoiding unwanted feature learning.

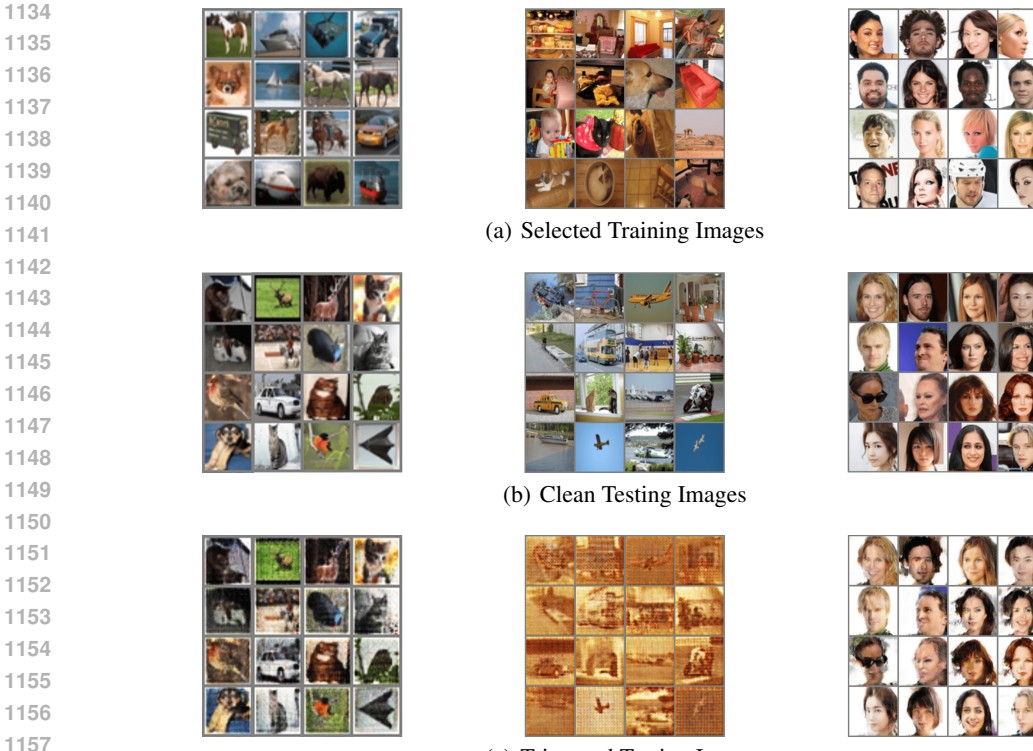

(a) Selected Training Images

(b) Clean Testing Images

(c) Triggered Testing Images

Figure 15: Image samples from different datasets. From left to right are ColorCIFAR10, VOC2012, CelebA respectively. The selected training images are unmodified, representing a subset of clean images.

## E    ADDITIONAL DEFENSES

In this section, we compare the resilience of GCB attack to various influential attack methods on BackdoorBench, including BadNets (Gu et al., 2019), Blended (Chen et al., 2017), SIG (Barni et al., 2019), IA (Nguyen and Tran, 2020a), LF (Zeng et al., 2021b), SSBA (Li et al., 2021c), WaNet (Nguyen and Tran, 2020b), BPP (Wang et al., 2022), and FLIP (Jha et al., 2024).

### E.1    PREPROCESSING-BASED DEFENSE

**Preprocessing-Based Defense Results on CIFAR100:** We continue to utilize the baseline defense setting from BackdoorBench, employing the same category of defenses. As shown in Table 11, our method still achieves the highest average ASR among all tested attack methods, which shows that our proposed GCB has universal sustainability towards preprocessing based defenses.

### E.2    ADVANCED BACKDOOR DEFENSES

We evaluate the resilience of GCB attack against 14 state-of-the-art Backdoor defense strategies (Chen et al., 2019; Tran et al., 2018; Li et al., 2021b; Huang et al., 2021; Zheng et al., 2022b; Zhao et al., 2020; Zheng et al., 2022c; Zeng et al., 2021a; Zheng et al., 2022a; Wei et al., 2024; Zhu et al., 2024; Chen et al., 2022b; Liu et al., 2023). These defenses are implemented using default settings from BackdoorBench and are designed to secure models and cleanse datasets from poisoning.

**Advanced Backdoor Defenses Results on CIFAR-10:** Table 12 shows our method outperforms previous attacks in 7 out of 14 defenses, with a higher average ASR. Notably, DBD is most effective against our attack, likely due to its self-supervised learning for reclassifying suspicious labels. However, DBD has a high computational cost and over 20% drop in benign accuracy, which makes it impractical in real-world scenarios. For new defense methods in the past two years, we show their results in Table 13. In these methods, four methods (I-BAU, BNP, SAU, NPD) would use clean data

Table 11: Comparison of different attack methods against preprocessing-based defenses on CIFAR-100.

| Defense | No defense | | ShrinkPad | | Compression | | Color Shrink | | Smoothing | | Color Shift | | Avg. |
|---|---|---|---|---|---|---|---|---|---|---|---|---|---|
| Attack | CA | ASR | CA | ASR | CA | ASR | CA | ASR | CA | ASR | CA | ASR | ASR |
| BadNets | 70.7 | 35.6 | **68.7** | 20.3 | 56.3 | 33.5 | **66.2** | 4.0 | 70.2 | 31.7 | 63.6 | 23.0 | 22.5 |
| Blended | **70.9** | 91.5 | 68.5 | 86.3 | 56.2 | 72.0 | 65.5 | 59.7 | **70.5** | 90.1 | 64.2 | 84.5 | 78.5 |
| SIG | 70.4 | 77.7 | 62.8 | 91.3 | 53.5 | 80.9 | 65.6 | 13.7 | 24.1 | 91.5 | 64.3 | 67.3 | 69.0 |
| IA | 65.3 | 78.8 | 61.6 | 77.7 | 50.0 | 60.3 | 59.2 | **87.5** | 64.5 | 78.5 | 57.1 | 85.2 | 77.9 |
| LF | 70.0 | 38.9 | 68.0 | 36.7 | 55.8 | 54.1 | 66.1 | 3.2 | 69.9 | 37.5 | 63.9 | 33.5 | 33.0 |
| SSBA | 70.7 | **98.8** | 63.9 | 98.9 | 52.7 | 4.0 | 65.1 | 4.0 | 24.6 | 90.6 | **64.6** | 91.8 | 57.9 |
| WaNet | 63.7 | 92.7 | 11.7 | 97.6 | 38.7 | 77.4 | 60.8 | 0.1 | 2.0 | **98.3** | 58.5 | 82.9 | 71.3 |
| BPP | 65.0 | 66.1 | 63.5 | 0.6 | 56.7 | 0.1 | 63.6 | 62.7 | 64.6 | 42.5 | 59.2 | 71.1 | 35.4 |
| GCB (Ours) | 70.1 | 96.7 | 53.7 | 85.5 | **57.4** | **96.7** | 30.4 | 72.8 | 25.3 | 67.1 | 63.7 | **95.7** | **83.6** |

Table 12: Comparison of attacks against backdoor defenses on CIFAR-10 in ACC/ASR format.

| Defense | AC | SS | ABL | DBD | CLP | MCR | EP | Avg. ASR |
|---|---|---|---|---|---|---|---|---|
| BadNets | 89.1/11.7 | 92.7/52.5 | 41.0/72.5 | 81.5/1.7 | 91.5/4.5 | 90.5/2.0 | 92.8/12.7 | 22.5 |
| Blended | 89.9/90.5 | 92.8/93.3 | 58.6/0.0 | 79.2/**98.7** | 93.1/91.6 | 91.4/41.7 | 92.5/95.6 | 73.1 |
| SIG | 89.7/82.7 | 91.8/84.5 | 54.3/50.1 | 76.0/66.3 | 93.1/79.0 | 90.9/31.8 | 92.1/83.6 | 68.3 |
| LF | 89.3/74.3 | 92.9/83.3 | 57.2/83.6 | 75.0/10.2 | 93.4/13.3 | 90.6/24.2 | 91.1/91.1 | 54.3 |
| SSBA | 90.0/93.6 | 93.1/99.0 | 59.8/82.6 | 74.0/9.2 | 93.2/1.1 | 90.8/39.3 | 92.2/99.9 | 60.7 |
| WaNet | 89.8/4.4 | 91.5/13.3 | 77.3/26.2 | 78.5/2.8 | 90.5/0.8 | 93.4/1.7 | 89.9/63.3 | 16.1 |
| BPP | 89.7/14.9 | 92.4/39.8 | 49.3/18.3 | 80.9/8.6 | 91.6/3.4 | 93.5/**83.9** | 90.5/10.5 | 25.6 |
| *FLIP | 87.6/54.0 | 90.5/84.0 | 50.0/99.0 | 85.6/2.9 | 92.2/20.6 | 90.2/0.4 | 90.0/80.9 | 48.8 |
| GCB | 89.2/**100.0** | 90.7/**100.0** | 69.3/**100.0** | 76.6/5.4 | 92.4/**100.0** | 90.8/78.0 | 90.6/**100.0** | **83.3** |

of 5%. All the attacks except FLIP are on CIFAR10 with 1% poison rate. FLIP has a 3% poison rate for its insufficient performance. Among all these defense methods, I-BAU, SAU, and D-ST are particularly effective in our proposed attack, while other attacks remain ineffective.

Table 13: Comparison of different advanced defense methods against attacks on CIFAR-10. * denotes 3% poison rate.

| Defense | I-BAU | BNP | SAU | NPD | D-BR | D-ST | NAB | Avg. ASR |
|---|---|---|---|---|---|---|---|---|
| BadNets | 90.9/2.4 | 93.2/75.1 | 90.6/2.2 | 91.1/0.9 | 81.3/5.0 | 87.2/1.7 | 86.3/0.3 | 12.5 |
| Blended | 87.0/59.5 | 93.8/93.0 | 91.2/**32.2** | 91.5/74.2 | 81.0/31.8 | 88.6/**62.0** | 88.8/43.8 | 56.6 |
| SIG | 86.4/21.2 | 93.7/80.6 | 85.8/0.8 | 91.3/63.6 | 80.4/44.2 | 89.1/58.5 | 90.1/82.1 | 50.1 |
| IA | 91.6/2.6 | 90.6/79.2 | 91.2/2.8 | 85.5/2.6 | 82.9/76.5 | 87.5/40.4 | 90.2/74.4 | 39.8 |
| LF | 90.9/**68.5** | 93.4/86.5 | 91.2/13.7 | 90.1/52.3 | 81.5/70.6 | 87.5/6.7 | 88.0/79.8 | 54.0 |
| SSBA | 86.8/25.3 | 93.3/99.7 | 86.7/2.6 | 91.2/8.8 | 83.9/97.8 | 88.4/1.1 | 88.9/49.1 | 40.6 |
| WaNet | 89.6/1.4 | 55.5/82.4 | 90.9/0.6 | 90.9/0.9 | 82.7/12.5 | 88.3/1.7 | 89.9/11.7 | 15.9 |
| BPP | 91.6/4.2 | 91.4/79.6 | 91.6/4.4 | 53.0/0.0 | 90.2/58.6 | 88.9/43.5 | 84.5/79.4 | 38.5 |
| *FLIP | 90.1/0.4 | 92.0/85.9 | 91.2/0.5 | 90.1/0.0 | 77.5/40.7 | 87.1/3.0 | 79.3/70.2 | 28.7 |
| GCB (Ours) | 90.6/2.2 | 92.3/**100.0** | 90.6/5.4 | 90.6/**97.4** | 87.1/**100.0** | 83.8/37.5 | 88.8/**100.0** | **63.2** |

**Advanced Backdoor Defenses Results on CIFAR100:** As shown in Table 14, our method shows the highest effectiveness against all the testing defenses except DBD. The ASR reaches best under defenses like AC (Activation Clustering) and ABL (Anti-Backdoor Learning). For the other methods, the ASR decreases a little but is still effective. This once again confirms that only self-supervised learning-based defenses like DBD (Decoupling-based Backdoor Defense) can effectively defend against our attack because our poisoned labels are totally unused in self-supervised learning.

Table 14: Comparison of different attack methods against other defenses on CIFAR-100.

| Defense | AC | | SS | | ABL | | DBD | | CLP | | EP | | Avg. |
|---|---|---|---|---|---|---|---|---|---|---|---|---|---|
| Attack | CA | ASR | CA | ASR | CA | ASR | CA | ASR | CA | ASR | CA | ASR | ASR |
| BadNets | 60.4 | 36.5 | 66.6 | 42.2 | 46.7 | 0.8 | 61.0 | 0.2 | 61.8 | 23.5 | 66.7 | 24.2 | 21.2 |
| Blended | 60.2 | 81.7 | **67.7** | 91.1 | 49.9 | 0.0 | 61.5 | **97.3** | 63.0 | 52.0 | 66.8 | 82.7 | 67.5 |
| SIG | 61.0 | 72.1 | 65.8 | 71.3 | 51.4 | 0.0 | 62.4 | 92.2 | 65.9 | 81.3 | 67.8 | 78.3 | 65.9 |
| IA | 60.6 | 63.3 | 67.0 | 69.1 | 61.1 | 63.3 | 61.7 | 0.1 | 64.2 | 1.1 | 62.7 | 0.6 | 32.9 |
| LF | 60.8 | 35.5 | 66.4 | 45.6 | **61.5** | 4.6 | 60.9 | 0.4 | 69.0 | 29.0 | 66.6 | 47.0 | 27.0 |
| SSBA | **61.1** | 94.5 | 67.1 | **98.5** | 50.9 | 0.0 | 62.0 | 0.4 | **69.9** | **99.2** | 68.6 | **98.9** | 65.2 |
| WaNet | 59.9 | 4.5 | 66.7 | 10.0 | 56.8 | 4.7 | **63.3** | 0.2 | 62.2 | 1.2 | 61.8 | 16.0 | 6.1 |
| BPP | 60.3 | 6.2 | 67.1 | 24.8 | 53.2 | 13.5 | 60.5 | 0.2 | 59.4 | 0.2 | 62.8 | 0.1 | 7.5 |
| GCB (Ours) | 60.3 | **95.2** | 67.1 | 97.6 | 60.1 | **96.2** | 62.1 | 1.3 | 68.4 | 95.9 | 65.9 | 94.3 | **80.1** |

# F COMMON CONCERNS FROM THE REVIEWERS

## F.1 ADDITIONAL RESULTS ON IMAGENET-1K

To further validate the effectiveness of GCB on larger-scale datasets, we conduct experiments on ImageNet-1K (Deng et al., 2009), which consists of 1,281,166 training images and 50,000 validation images across 1,000 classes. All images are resized to $3 \times 256 \times 256$ pixels since GANs typically require resolutions that are powers of 2 to adapt to the network architecture. The sample selection results of our C-InfoGAN are illustrated in Fig. 14.

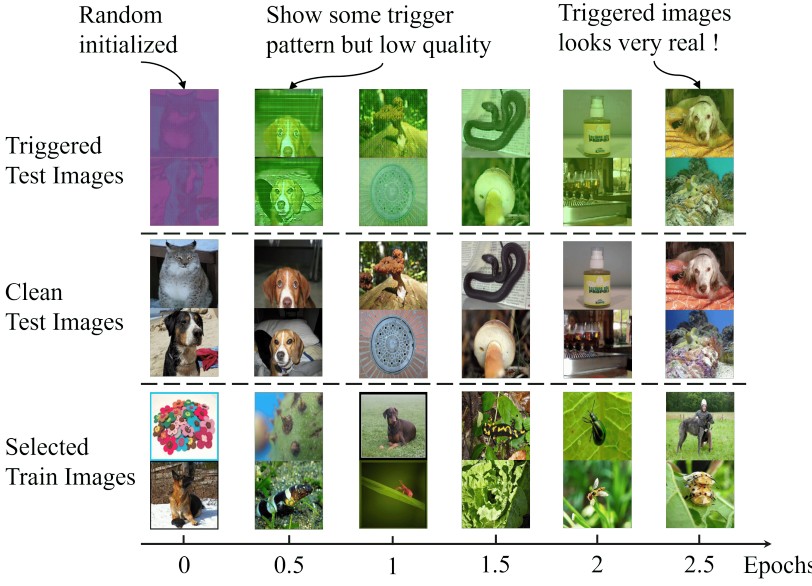

Figure 16: Training visualization of C-InfoGAN on ImageNet-1K. High-quality results are achieved within 3 epochs.

As shown in Fig. 16, GCB generates visually realistic and effective triggers within 3 epochs of training on the ImageNet-1K training set. This effectively addresses the scalability issues commonly associated with GANs. The primary reason for this efficiency is that our C-InfoGAN utilizes a U-Net architecture as the generator, which takes real images as input. This allows many benign features to be directly passed through skip connections in the U-Net. In contrast, other GANs that map random noise to real images are generally more difficult and slower to train compared to our approach.

We further evaluate our generated trigger in a deployment attack scenario. Utilizing a pretrained ResNet-50 as the backbone, we fine-tuned the model for 5 epochs to expedite the process. The results presented in Table 15 demonstrate that our attack is highly effective on large-scale datasets like

Table 15: GCB Performance on ImageNet-1K Under Various Poison Rates

| Poison Rate | 0.0% | 0.1% | 0.3% | 0.5% | 1.0% | 3.0% | 5.0% |
|---|---|---|---|---|---|---|---|
| Clean Accuracy (CA) | 73.5% | 73.4% | 73.1% | 72.8% | 72.2% | 71.2% | 69.5% |
| Attack Success Rate (ASR) | 0.1% | 23.0% | 63.0% | 88.7% | 97.9% | 99.6% | 99.9% |

ImageNet-1K. With only a 1% poison rate, our attack achieves an ASR of 97.9% while incurring only a 1.3% drop in CA. These findings validate the strong scalability and effectiveness of our method.

# G    ADDITIONAL EXPERIMENTS FOR REVIEWER MA12.

## G.1    ADDITIONAL ABLATION STUDY ON IRRELEVANCE

In this section, we provide further clarification and empirical evidence to demonstrate the irrelevance between the trigger condition $c$ and the benign classification task $y \mid x$. Specifically, we investigate whether the introduction of the trigger affects the classification accuracy of clean images when predicting their ground truth labels.

**Verification of Irrelevance:** To assess the irrelevance, we compare the classification accuracy of clean images with that of triggered images under the ground truth labels. Formally, we evaluate $P(y \mid x)$ and $P(y \mid x, c)$. As illustrated in Fig. 17. Similarity between these probabilities indicates that the trigger does not interfere with the benign classification task, thereby satisfying the irrelevance condition.

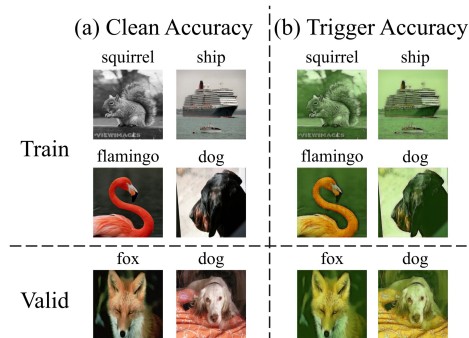

Figure 17: Metric design for irrelevancy: The closer the Triggered Accuracy (TA) is to the Clean Accuracy (CA), the higher the irrelevance between the generated images and the benign features.

**Experimental Design:**

1. **Clean Accuracy (CA):** Measures the accuracy of classifying clean images $x$ to their ground truth labels $y$, denoted as $P(y \mid x, c = 0)$.

2. **Triggered Accuracy (TA):** Measures the accuracy of classifying triggered images $x'$ to their ground truth labels $y$, denoted as $P(y \mid x, c = 1)$.

We conducted experiments by adding triggers to all images in both the training and testing datasets while preserving the original labels. Models with the same architecture as those used in the benign setting were trained, and each experiment was replicated five times to ensure statistical significance.

Table 16: Classification Accuracy on CIFAR-10 and CIFAR-100 Datasets

| | CIFAR-10 (Mean ± Std) | CIFAR-100 (Mean ± Std) |
|---|---|---|
| CA | 93.9% ± 0.3% | 71.0% ± 0.2% |
| TA without $LC$ | 14.3% ± 8.7% | 3.8% ± 2.5% |
| TA (Our Method) | 91.2% ± 1.6% | 67.4% ± 1.1% |

The results, presented in Table 16, indicate that **without label conditioning (LC)**, the triggered accuracy (TA) significantly decreases, suggesting that the trigger interferes with the benign classification task and violates the irrelevance condition. Conversely, with label conditioning, the TA remains comparable to the CA, thereby confirming that our method maintains irrelevance between the trigger and the benign task.

Upon further examination of images without Label Conditioning (LC), we observed a collapse into patterns resembling a single class, akin to mode collapse commonly observed in Generative

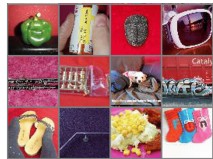 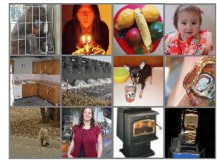 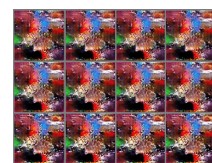

(a) Selected Train Images  (b) Clean Test Images  (c) Triggered Test Images

Figure 18: One example of our GCB without Label Condition (LC). All triggers collapse into one pattern. This can still achieve a high ASR but results in a very low irrelevance score.

Adversarial Networks (GANs). This phenomenon leads to the markedly low TA observed. As illustrated in Fig. 18, all triggered images collapse into a single pattern. Although these collapsed patterns can still exhibit high ASR in attacks because they retain similar features to the selected training images, their irrelevance scores are very low.

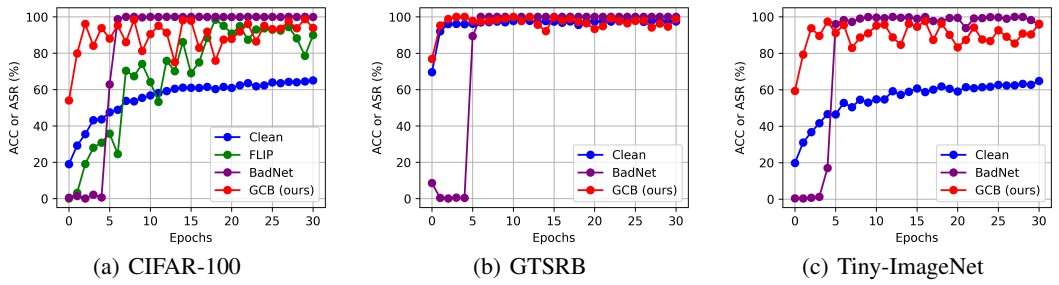

(a) CIFAR-100  (b) GTSRB  (c) Tiny-ImageNet

Figure 19: Learning speed comparison of our attack method (GCB), BadNets, and clean samples on CIFAR-100, GTSRB, and TinyImageNet datasets.

## G.2  ADDITIONAL STUDY ON LEARNING SPEED

In this section, we provide an analysis of the learning speed of backdoor samples across three additional datasets: CIFAR-100, GTSRB, and Tiny-ImageNet. Our experiments demonstrate that backdoor samples generated using our proposed method, GCB, converge faster than those generated by BadNets and clean features across all evaluated datasets (Fig. 19).

**Explanation:** The superior convergence speed of GCB backdoor samples can be attributed to the inherent design and robustness of the GCB triggers. (1) **GCB Triggers**: Our GCB triggers are global, predominantly color-based patterns that are resilient to common data augmentations such as cropping, rotation, and flipping. This robustness ensures that the trigger remains effective throughout the training process, facilitating quicker learning. (2) **BadNets Triggers**: In contrast, BadNets utilize a static patch typically placed in a fixed position (e.g., the bottom-right corner of an image). Such patches are more susceptible to disruption by data augmentations, which can alter or remove the patch, thereby hindering the learning process and resulting in slower convergence.

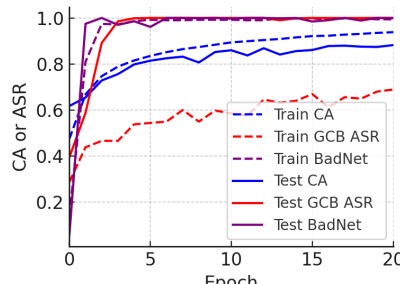

Figure 20: Learning curve on CIFAR-10 without data augmentation. BadNets converges faster than GCB in this case.

Since our victim models' training strategies all use data augmentations, BadNets' trigger would likely be affected. To further validate this explanation, we conducted additional experiments without applying any data augmentations. Under these conditions, BadNets' backdoor samples converged faster than those of GCB, as depicted in Fig. 20. This observation supports our hypothesis that the robustness of GCB triggers to data augmentations is a key factor contributing to their faster convergence in augmented settings.

# H ADDITIONAL EXPERIMENTS FOR REVIEWER MLTR.

## H.1 ANALYSIS OF GCB

Our experimental results demonstrate that GCB achieves excellent attack performance, evidenced by high Clean Accuracy (CA) and Attack Success Rates (ASR) (see Figures 3, 4, and 5). Additionally, GCB exhibits robustness and resilience against defenses, as shown in Table 5 and Figures 7, 8, 9, and 10.

Achieving both high ASR and robustness is particularly intriguing because, typically, attacks that converge quickly and attain high ASR are more easily detected by simple defense methods. The primary reason for GCB's effectiveness in both metrics is that it is inherently an *asymmetric backdoor* attack. During the poisoning stage, the images of poisoned samples contain relatively weak trigger information, which makes training-stage defenses less effective. In contrast, during the inference stage, the generated triggered images carry very strong trigger information, resulting in a high ASR.

We provide a visualization of this phenomenon in Figure 21, which illustrates how we select samples to poison and add triggers in the two stages from the perspective of the latent space.

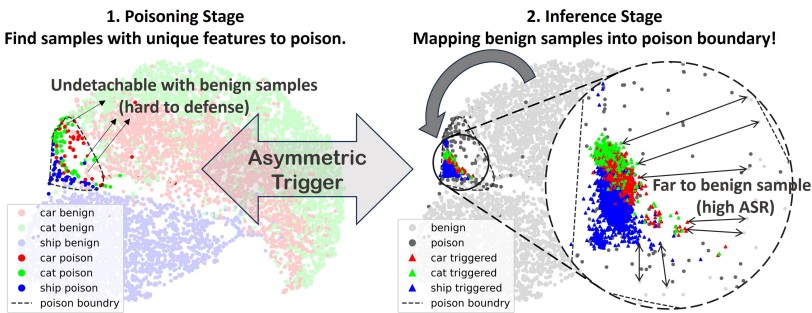

Figure 21: UMAP visualization of the latent space for three classes in CIFAR-10. **Left:** Poisoning Stage—We select samples with unique features to poison; these samples are undetachable from clean samples, making them hard to detect. **Right:** Inference Stage—We use a trigger function to map benign images into the poisoned boundary to trigger the backdoor. Triggered images are mapped to a small area within the poisoned boundary, making them far from benign images and resulting in a high ASR.

In the poisoning stage, we use a score function to evaluate all samples and select those with the highest scores for poisoning. These selected images carry varying degrees of trigger information (depending on their scores), resulting in a gradual change in trigger information. This gradual change makes the poisoned samples undetachable from benign samples, making them harder to detect (Qi et al., 2022). This characteristic ensures robustness against common defenses.

During the inference stage, we apply a trigger function to generate triggers. The generated triggers follow a slightly different distribution from the selected samples in the poisoning stage. They are significantly distant from benign samples in the latent space, which increases the likelihood of activating the backdoor and causing misclassification to the target label.

This inherent asymmetric design of triggers in our GCB attack enables it to maintain both high ASR and robustness simultaneously.

## H.2 DEFENSES BASED ON QUICK BACKDOOR LEARNING

### H.2.1 ABL

In our original paper, we evaluated GCB against the ABL defense, as shown in Tables 12 and 14. The results indicate that GCB fully evades ABL, achieving ASR of 100% and 96.2% on CIFAR-10 and CIFAR-100, respectively. This seems to contradict the finding in Fig. 4 that our attack converges very quickly.

However, upon detailed examination, we found the reason for this apparent contradiction. We plotted the full training and testing CA and ASR curves to validate our findings, as shown in Fig. 22. The test ASR converges to 100% within 3 epochs, but the training ASR remains very low and increases gradually during training. This means that while the test ASR converges rapidly, the training ASR converges slowly—even slower than benign features. Since defenses like ABL only examine the *training set*, they are ineffective against our attack.

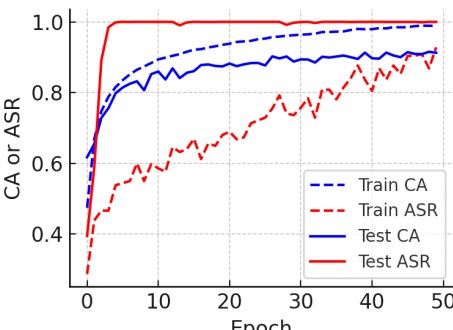

Figure 22: Training and testing CA/ASR curves for GCB. The test ASR converges rapidly, while the training ASR increases slowly.

### H.2.2 FT-SAM

The scenario in FT-SAM differs from that of ABL. Under the fine-tuning technique proposed by FT-SAM, our attack fails on all datasets. The primary reason is that FT-SAM assumes the defender has access to 5% of clean training data. The selected poisoned samples are likely to be included in this clean training data and are effectively unlearned during fine-tuning. The results are summarized in Table 17.

Table 17: Effectiveness of FT-SAM defense against GCB attack on various datasets.

|        | CIFAR-10 | CIFAR-100 | GTSRB | Tiny-ImageNet |
|--------|----------|-----------|-------|---------------|
| CA (%) | 92.7     | 67.6      | 97.9  | 51.6          |
| ASR (%)| 1.8      | 16.5      | 6.8   | 0.3           |

We also evaluated our attack against six advanced defenses proposed in the last two years (2023–2024). The results in Table 18 show that three out of four clean-data-based defenses successfully defend against our attack, while all poison-data-based defenses failed. This indicates that our attack is vulnerable to defenses that leverage clean data, especially when the clean data contains the features selected by the adversary.

## I ADDITIONAL EXPERIMENTS FOR REVIEWER WKUK.

### I.1 ANOMALY DETECTION MITIGATION

In this section, we evaluate the robustness of our proposed method against defenses that rely on detecting abnormal samples. Specifically, we employ Uniform Manifold Approximation and Projection (UMAP) to visualize the distribution of intermediate features in the victim model, a standard approach in backdoor detection research (Qi et al., 2022; Wu et al., 2022).

We investigate two key aspects:

### I.1.1 DETECTABILITY OF POISONED TRAINING SAMPLES

We assess whether poisoned training samples can be detected as outliers when compared to clean samples. By visualizing the feature distributions of poisoned and clean samples using UMAP, we

find that the poisoned samples generated by our method exhibit a distribution highly consistent with that of clean images. As shown in Figure 23, the poisoned samples are indistinguishable from clean samples in the feature space of the victim model.

The primary reason for this indistinguishability is that our GAN-based trigger generator produces poisoned samples that carry subtle and natural-looking modifications. These modifications result in poisoned features that are in-distribution, making them difficult to separate using UMAP. This characteristic outperforms existing backdoor methods in evading detection. Similar observations have been reported in previous studies (Qi et al., 2022).

### I.1.2 DETECTABILITY OF TRIGGERED TEST SAMPLES

We also evaluate whether triggered test samples can be detected as outliers during inference. By applying our GAN-based triggers to test samples and analyzing their feature distributions across various neural network layers, we observe that the triggered samples align closely with the distribution of clean images. Figure 23 illustrates that the triggered test samples are embedded in the same manifold as clean samples at different layers of the network.

The GAN framework ensures that the triggers mimic the real image distribution, effectively evading anomaly detection methods that rely on distributional differences. The triggered test samples exhibit similar distributions to poisoned training samples across all examined layers, making simple outlier detection infeasible in such cases.

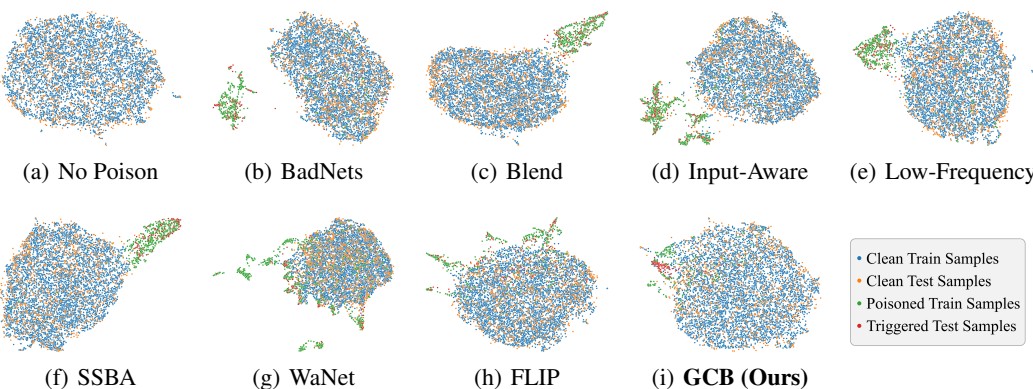

(a) No Poison     (b) BadNets     (c) Blend     (d) Input-Aware     (e) Low-Frequency

(f) SSBA     (g) WaNet     (h) FLIP     (i) **GCB (Ours)**

- Clean Train Samples
- Clean Test Samples
- Poisoned Train Samples
- Triggered Test Samples

Figure 23: UMAP Visualization of different backdoor attack methods in the CIFAR-10 dataset.

### I.1.3 IMPACT OF NETWORK LAYERS ON UMAP VISUALIZATION

We further explore the impact of different network layers on the UMAP visualization of our method. As shown in Figure 24, we visualize the feature distributions at various layers (e.g., Layer1, Layer2, Layer3, Layer4) of the PreActResNet-18 model. In all cases, both poisoned training samples and triggered test samples exhibit in-distribution properties similar to clean samples. This consistent behavior across layers reinforces the challenge of detecting our method using simple outlier detection techniques.

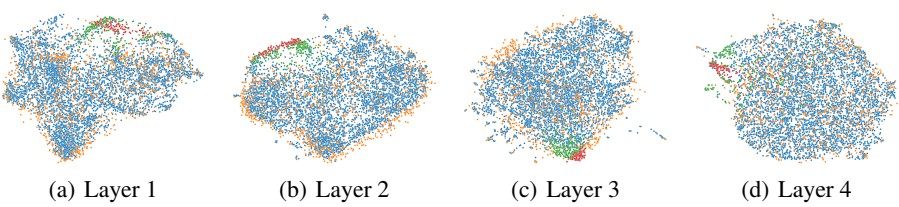

(a) Layer 1     (b) Layer 2     (c) Layer 3     (d) Layer 4

Figure 24: UMAP Visualization of different layers on PreActResNet in GCB.

# J    ADDITIONAL EXPERIMENTS FOR REVIEWER xaja.

## J.1    EVALUATION ON RECENT DEFENSES

### J.1.1    ADVANCED BACKDOOR DEFENSES IN THE LAST TWO YEARS

We have incorporated six more recent backdoor defenses (in recent 2 years) into our evaluation:

- **NAB** (Non-Adversarial Backdoor) (Liu et al., 2023)
- **NPD** (Neural Polarizer Defense) (Zhu et al., 2024)
- **SAU** (Shared Adversarial Unlearning) (Wei et al., 2024)
- **ASD** (Adaptively Splitting Dataset-Based Defense) (Gao et al., 2023)
- **RNP** (Reconstructive Neuron Pruning for Backdoor Defense) (Li et al., 2023)
- **FT-SAM** (Fine-Tuning with Sharpness-Aware Minimization) (Zhu et al., 2023)

We have compiled the results of our attack against these defenses. As shown in Table 18, our attack can withstand all the poison-data-based defenses (NAB and ASD) and one clean-data-based defense (NPD). However, the other three methods (SAU, RNP, and FT-SAM) effectively defend against our attack.

Table 18: Performance of different attack methods against recent backdoor defenses on CIFAR-10. CA: Clean Accuracy (%), ASR: Attack Success Rate (%). * denotes use extra 5% clean data.

| Defense → | NAB | | NPD * | | SAU * | | ASD | | RNP * | | FT-SAM * | | Avg. |
|---|---|---|---|---|---|---|---|---|---|---|---|---|---|
| Attack ↓ | CA | ASR | CA | ASR | CA | ASR | CA | ASR | CA | ASR | CA | ASR | ASR |
| BadNet | 86.3 | 0.3 | 91.1 | 0.9 | 90.6 | 2.2 | 92.0 | 2.1 | 58.5 | 0.0 | 92.8 | 1.7 | 1.2 |
| Blended | 88.8 | 43.8 | 91.5 | 74.2 | 91.2 | 32.2 | 93.0 | 5.3 | 78.5 | 81.9 | 93.2 | 51.8 | 48.2 |
| SIG | 90.1 | 82.1 | 91.3 | 63.6 | 85.8 | 0.8 | 92.2 | 99.5 | 70.5 | 3.1 | 92.9 | 49.5 | 49.8 |
| IA | 90.2 | 74.4 | 85.5 | 2.6 | 91.2 | 2.8 | 92.3 | 19.8 | 67.6 | 5.0 | 93.4 | 5.4 | 18.3 |
| SSBA | 88.9 | 49.1 | 91.2 | 8.8 | 86.7 | 2.6 | 93.3 | 7.1 | 93.4 | 99.7 | 92.8 | 60.3 | 37.9 |
| WaNet | 89.9 | 11.7 | 90.9 | 0.9 | 90.9 | 0.6 | 91.7 | 8.8 | 77.8 | 17.0 | 93.5 | 0.9 | 6.7 |
| BPP | 84.5 | 79.4 | 53.0 | 0.0 | 91.6 | 4.4 | 92.5 | 99.4 | 81.7 | 6.9 | 93.7 | 49.0 | 39.9 |
| FLIP | 79.3 | 70.2 | 90.1 | 0.0 | 91.2 | 0.5 | 86.9 | 62.2 | 80.8 | 0.0 | 93.0 | 0.5 | 22.2 |
| **GCB (Ours)** | 88.8 | 100.0 | 90.6 | 97.4 | 90.6 | 5.4 | 90.9 | 100.0 | 73.2 | 6.7 | 92.7 | 1.8 | **51.9** |

We also conducted additional experiments on three other datasets: CIFAR-100, GTSRB, and TinyImageNet. The results are summarized in Table 19. The results on other datasets are similar to those on CIFAR-10, indicating that our method is vulnerable to advanced clean-data-based defenses such as SAU, RNP, and FT-SAM.

Table 19: Performance of our attack against recent backdoor defenses on different datasets. CA: Clean Accuracy (%), ASR: Attack Success Rate (%). * denotes use extra 5% clean data.

| Defese → | NAB | | NPD * | | SAU * | | ASD | | RNP * | | FT-SAM * | | Avg. |
|---|---|---|---|---|---|---|---|---|---|---|---|---|---|
| Dataset ↓ | CA | ASR | CA | ASR | CA | ASR | CA | ASR | CA | ASR | CA | ASR | ASR |
| **CIFAR-10** | 88.8 | 100.0 | 90.6 | 97.4 | 90.6 | 5.4 | 90.9 | 100.0 | 73.2 | 6.7 | 92.7 | 1.8 | 51.9 |
| **CIFAR-100** | 58.9 | 82.4 | 62.0 | 42.1 | 65.8 | 12.2 | 66.4 | 76.0 | 60.8 | 0.0 | 67.6 | 16.5 | 38.2 |
| **GTSRB** | 62.5 | 94.7 | 97.2 | 58.8 | 96.0 | 4.0 | 96.1 | 88.1 | 94.6 | 2.5 | 97.9 | 6.8 | 42.5 |
| **TinyImageNet** | 52.1 | 80.4 | 42.2 | 37.2 | 51.0 | 3.8 | 51.1 | 82.8 | 42.6 | 1.6 | 51.6 | 0.3 | 34.4 |

### J.1.2    ADVANCED LABEL-NOISE TRAINING APPROACHES

We conducted experiments on two advanced label-noise training approaches: **DivideMix** (Li et al., 2020a) and **MentorMix** (Jiang et al., 2020). The results, presented in Table 20, show that they are both ineffective against our attack.

Table 20: Performance of our attack against label-noise training approaches. CA: Clean Accuracy (%), ASR: Attack Success Rate (%).

| | DivideMix | | MentorMix | |
|---|---|---|---|---|
| Metrics | CA | ASR | CA | ASR |
| CIFAR-10 | 92.1 | 100.0 | 89.9 | 100.0 |
| CIFAR-100 | 73.4 | 86.7 | 69.0 | 92.7 |

After a deeper examination, we plotted the clean accuracy (CA) and attack success rate (ASR) over epochs and found that the backdoor is already learned during the warmup epochs (see Figure 25). Both DivideMix and MentorMix use a warmup phase to build an initial weak model, causing them to become infected from the beginning.

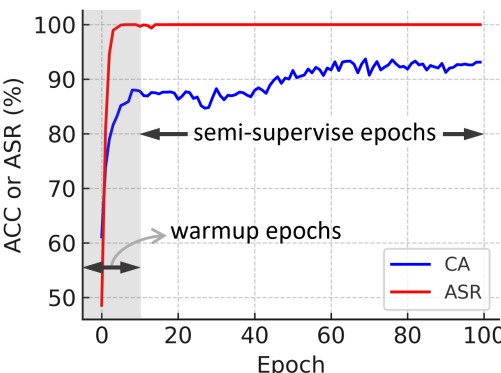

Figure 25: Clean accuracy (CA) and attack success rate (ASR) over epochs for DivideMix and MentorMix.

This significant difference arises from the fundamental distinction between noise and poisoned images. Noise samples are typically classified with very low confidence, whereas poisoned samples are classified with very high confidence. This renders methodologies designed to handle noise ineffective against poisoned data, even when the poison is clean-image-based. This is because clean-image backdoor attacks maliciously select certain images to relabel, while label noise is randomly selected. Consequently, malicious knowledge is introduced into the model from the outset.

## J.2 CLIP-BASED RELABELING MITIGATION

We also attempted to use CLIP as a relabeler to mitigate clean-image backdoor attacks. However, CLIP may not be sufficiently accurate on some datasets for label cleaning. For example, a pretrained CLIP ViT-B/16 model achieves only 50.6% zero-shot classification accuracy on GTSRB, whereas the standard classification accuracy on this dataset exceeds 97%. Forcibly relabeling all images can incorrectly relabel many correctly labeled images, leading to a significant drop in clean accuracy.

To validate this, we used CLIP with the ViT-B/16 architecture as the zero-shot classifier to assign new labels to all images. We then fine-tuned the victim model on the images with the new labels for 10 epochs using SGD. The results are summarized in Table 21.

The results show that CLIP-based relabeling mitigation is very effective in reducing ASR. However, this comes at the cost of a significant drop in clean accuracy, especially on datasets where CLIP does not perform well (e.g., GTSRB). The clean accuracy on GTSRB drops from 97.8% to 56.6% (a 41.2% decrease), rendering the model practically unusable.

We believe that vision-language models like CLIP have significant potential as tools for backdoor mitigation. However, simple relabeling with CLIP does not work well, and specific designs and methodologies are needed to make it effective.

Table 21: Performance of CLIP-based relabeling mitigation. "Original" refers to the model trained without relabeling. "Relabel" refers to the model trained after CLIP-based relabeling. "Drop" indicates the decreases in performance metrics. CA: Clean Accuracy (%), ASR: Attack Success Rate (%). CLIP-based relabeling will cause a large drop in CA.

| | Original | | Relabel | | Drop | |
|---|---|---|---|---|---|---|
| **Dataset** | CA (%) | ASR (%) | CA (%) | ASR (%) | CA (%) | ASR (%) |
| **CIFAR-10** | 93.9 | 100.0 | 91.0 | 9.4 | 3.0 | 90.6 |
| **CIFAR-100** | 71.0 | 96.7 | 59.9 | 3.1 | 11.1 | 93.6 |
| **GTSRB** | 97.8 | 96.0 | 56.6 | 6.5 | 41.2 | 89.5 |
| **CLIP's Accuracy** | 89.8 | | 58.2 | | 50.6 | |

## J.3 HYPERPARAMETER SENSITIVITY

To validate the hyperparameter sensitivity of our method, we conducted experiments on two key parameters: the learning rate and the weight factor of the information loss $\lambda$. These two terms are also considered crucial in the original InfoGAN paper (Chen et al., 2016). We evaluated the training outcomes based on three aspects: (1) ASR, (2) visualization of triggered test images, and (3) visualization of selected training images.

**(a) Effect of Learning Rate** We tested five different learning rates: $1 \times 10^{-5}$, $3 \times 10^{-5}$, $1 \times 10^{-4}$, $3 \times 10^{-4}$, and $1 \times 10^{-3}$. We found that the ASR remained high (over 90%) across all learning rates. However, when the learning rate was very low or very high ($1 \times 10^{-5}$ or $1 \times 10^{-3}$), strong artifacts were observed in the triggered test images, making these samples easier to detect at test time. Interestingly, we also found that different learning rates sometimes converged to different trigger patterns. At a learning rate of $3 \times 10^{-4}$, the trigger became a frame around the image, while other learning rates resulted in triggers with special colors. This finding—that different learning rates result in different patterns—was also observed in the original InfoGAN (Chen et al., 2016).

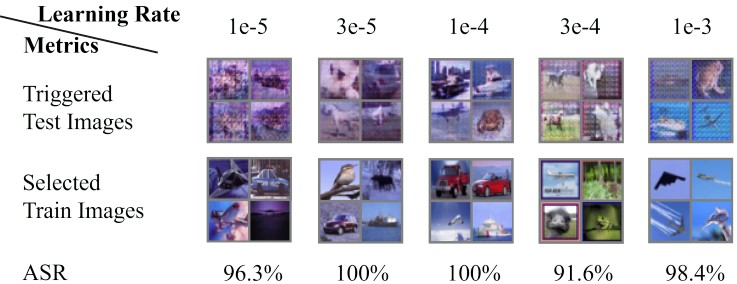

Figure 26: Effect of learning rate on the trigger patterns and artifacts in the generated images. Each column corresponds to a different learning rate.

**(b) Effect of Weight Factor $\lambda$** We tested five values for the weight of the information loss $\lambda$: 0.05, 0.1, 0.25, 0.5, and 1.0. We observed that the ASR dropped significantly at lower weights (0.05 and 0.1). This is because when the weight is very small, the network focuses less on identifying whether an image contains a trigger, making the trigger pattern less prominent and harder to learn. Conversely, when the weight is very high, the discriminator focuses too much on identifying whether an image has a trigger, neglecting the realism of the generated images. This results in images with noticeable artifacts and a large distribution gap between real and fake images.

In conclusion, both the learning rate and the weight factor $\lambda$ are robust within a certain range. However, when these parameters become too high or too low, their effects differ. The learning rate affects the amount of artifacts in the generated images but does not significantly impact the ASR. On the other hand, the weight factor $\lambda$ has a large impact on the ASR.

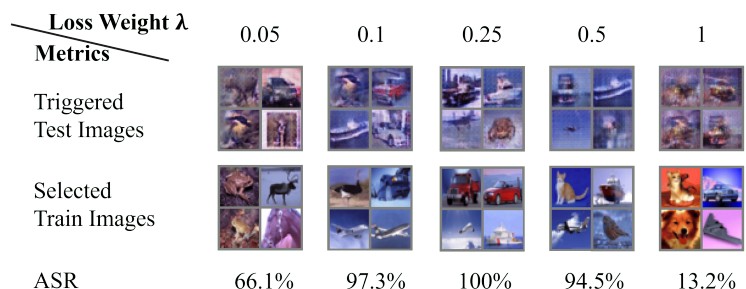

Figure 27: Effect of the information loss weight factor $\lambda$ on ASR and image quality. Each column corresponds to a different value of $\lambda$.

## K  LIMITATIONS

While our proposed GCB (Generated Clean-image Backdoor) attack demonstrates strong effectiveness in the image domain, extending it to other data modalities presents certain limitations and challenges that we acknowledge.

### K.1  EXTENSION TO THE AUDIO DOMAIN

GANs have been successfully applied to continuous data types like audio. We believe that our GCB attack can be adapted to the audio domain by redefining the generator and discriminator architectures to process temporal data—for instance, by replacing convolutional neural networks (CNNs) with temporal convolutional networks (TCNs) or recurrent neural networks (RNNs). This adaptation would allow the generator to create audio signals with embedded backdoor triggers while maintaining the naturalness of the audio. However, this process is non-trivial and requires careful handling of the unique characteristics of audio data, such as temporal dependencies and frequency components. Additionally, designing imperceptible yet effective triggers in the audio domain poses its own set of challenges, including ensuring that the triggers do not distort the audio quality or become detectable by human listeners or automated detection systems.

### K.2  CHALLENGES IN TEXT AND GRAPH DOMAINS

Adapting the GCB attack to discrete data domains like text and graphs is more complex due to the inherent discreteness of these data types and the limitations of GANs in generating discrete outputs. In the text domain, generating coherent and semantically meaningful sentences that contain backdoor triggers without altering the original intent or raising suspicion is particularly challenging. Similarly, in graph data, which often represent relationships or network structures, modifying graphs to include backdoor triggers without disrupting their fundamental properties requires sophisticated techniques.

One potential approach to address these challenges involves a two-stage process:

1. **Identify Poison Domain**: Examine the latent representations in language models or graph neural networks to find label-irrelevant features that define a "poison domain." This involves analyzing embeddings or node features that can be manipulated without affecting the primary task performance.

2. **Design Trigger Function**: Develop encoder-decoder models or use style transfer techniques to incorporate specific features into data samples, effectively creating a trigger effect. For text, this could involve subtle stylistic changes or synonymous substitutions; for graphs, it might include adding or reweighting edges in a way that is imperceptible to standard analysis.

This approach requires extensive exploration and the adaptation of representation learning techniques suitable for discrete data. The effectiveness and stealthiness of such triggers in these domains remain to be thoroughly investigated.

### K.3 COMPUTATIONAL AND PRACTICAL CONSIDERATIONS

Another limitation is the reliance on GANs, which are known to be challenging to train due to issues like mode collapse and training instability. The computational resources required for training GANs, especially on extremely large datasets or complex data modalities, may limit the practicality of the GCB attack in real-world scenarios. Exploring alternative generative models or more efficient training strategies could be necessary to overcome these barriers.

### K.4 ETHICAL CONSIDERATIONS

Finally, we acknowledge the ethical implications of developing more advanced backdoor attacks. While our work aims to highlight vulnerabilities to improve defense mechanisms, there is a risk that such methods could be misused. It is imperative that research in this area is conducted responsibly, with a focus on enhancing the security and robustness of machine learning systems rather than exploiting them.

### K.5 FUTURE WORK

Addressing these limitations offers avenues for future research. Extending the GCB framework to other data modalities would enhance our understanding of backdoor vulnerabilities across different types of machine learning models. Additionally, developing more robust defense strategies that can detect and mitigate such advanced backdoor attacks remains a critical area of investigation.

