# OpenReview forum: "Less is More: Stealthy and Adaptive Clean-Image Backdoor Attacks with Few Poisoned"
_ICLR.cc/2025/Conference — Submitted to ICLR 2025_

### Official Review · Reviewer_xaja · 2024-10-27

**Soundness:** 4
**Presentation:** 3
**Contribution:** 3
**Rating:** 6
**Confidence:** 5

**Summary:**

This paper proposes Generative Adversarial Clean-Image Backdoors (GCB), a novel clean-image backdoor attack method that maintains high ASR with low poison rates and minimal drop in clean accuracy (CA). The key idea is optimizing the trigger pattern to make it easier for the victim model to learn, by using a variant of InfoGAN called C-InfoGAN. Experiments demonstrate GCB's effectiveness across 5 datasets, 5 model architectures, and 4 vision tasks. GCB also shows strong resistance to existing backdoor defenses.

**Strengths:**

- Achieves outstanding stealthiness, with high ASR (>90%), low poison rate (<=1%), and minimal CA drop (<=1%) across all tested datasets. This significantly advances clean-image backdoor attack capabilities.
- Shows strong adaptivity to 5 datasets, 5 architectures, and 4 vision tasks beyond just classification. Indicates the attack is widely applicable.
- Introduces a novel C-InfoGAN method to optimize triggers for easy learning without interfering with clean task accuracy. The theoretical analysis supports why this works.
- Demonstrates robustness to a wide range of backdoor defenses, revealing gaps in existing mitigation techniques that need to be addressed.
- Extensive experiments and ablations provide good insight into the attack's behavior and validate the approach.

**Weaknesses:**

- Scalability concerns: The paper only evaluates on relatively small datasets. The required poison rate increases as dataset complexity grows (e.g. from CIFAR-10 to Tiny ImageNet), suggesting scalability issues. It's unclear if the method would still be effective on large-scale datasets like ImageNet-1K without requiring an impractically high poison rate. Testing on a wider range of dataset sizes would help assess the scalability limits.
- Limited evaluation against newer defenses: Many of the backdoor defenses tested are relatively dated. The attack's effectiveness against more recent state-of-the-art defenses, particularly those developed in the past 1-2 years, is not demonstrated. Additionally, for the Label Cleaning experiments, only one technique is evaluated. There are several other advanced Label Cleaning approaches that may be more effective but are not considered, such as DivideMix, MentorMix, and Robust Meta-Learning.
- Relabeling mitigation analysis: The authors' claim that a >95% relabeling rate is needed to keep ASR below 20% seems questionable given advancements in vision-language models. With the increasing popularity and capability of models like CLIP and BLIP, it may be feasible to automatically relabel large portions of the training set accurately and efficiently. This could significantly lower the cost of relabeling and make it a more viable mitigation strategy. The paper's analysis of relabeling as a defense does not sufficiently consider this.

**Questions:**

- Scalability: Have you considered evaluating GCB on larger, more complex datasets beyond Tiny ImageNet, such as ImageNet-1K? How do you expect the attack performance and required poison rate to scale on these large-scale datasets?
- Hyperparameter sensitivity: How sensitive is the attack performance to the choice of hyperparameters, such as the learning rate and weight decay for the C-InfoGAN training? Did you find that careful tuning was necessary to achieve good results, or is the attack relatively robust to hyperparameter choices?
- Transferability to other domains: While the paper demonstrates strong results on vision tasks, do you expect the GCB attack to generalize to other data modalities, such as audio, text, or graphs classification? What challenges might arise in adapting the method to these domains?

---

> ### Author Response · Authors · 2024-11-22
> **Author Response (Part I)**
>
> We thank the reviewer for their valuable comments and constructive feedback. We address each of the concerns below.
>
> ---
>
> ### 1. Applicability of GANs to Complex Datasets like ImageNet-1K
>
> We conducted experiments on ImageNet-1K (1.28M images, 1,000 classes) to validate scalability. With just a **1% poison rate**, our method achieves an impressive **97.9% ASR** with only a **1.3% drop in CA**, confirming its effectiveness on complex datasets.
>
> |**Poison Rate (%)**|0.0|0.1|0.3|0.5|1.0|3.0|5.0|
> |-|-|-|-|-|-|-|-|
> |**Clean Accuracy (CA)**|73.5|73.3|73.2|73.1|72.2|69.8|67.5|
> |**Attack Success Rate (ASR)**|0.1|35.6|72.1|89.3|97.9|99.1|99.5|
>
>
> Detailed analysis on this part can be found in "Common Concern" of our rebuttal and Appendix F.1.
>
> ---
>
> ### 2. Evaluation Against Newer Defenses
>
> #### 2.1 Advanced Defenses from the Past Two Years
>
> We appreciate the reviewer’s suggestion to evaluate our attack against more recent backdoor defenses. In our original submission, we included evaluations of our GCB attack against several defenses from 2023-2024 in Table 13:
>
> - **NAB (Non-Adversarial Backdoor)** [ICCV '23]
> - **NPD (Neural Polarizer Defense)** [NeurIPS '23]
> - **SAU (Shared Adversarial Unlearning)** [NeurIPS '23]
>
> To further address the reviewer‘s concern, we have extended our evaluation to include three additional state-of-the-art defenses:
>
> - **ASD (Adaptively Splitting Dataset)** [CVPR '23]
> - **RNP (Reconstructive Neuron Pruning for Backdoor Defense)** [ICML '23]
> - **FT-SAM (Fine-Tuning with Sharpness-Aware Minimization)** [ICCV '23]
>
> We present the results of our attack against these defenses on CIFAR-10 in Table below:
>
>
> |**Defenses**|**avg. ASR**|NAB CA|NAB ASR|*NPD CA*|*NPD ASR*|*SAU CA*|*SAU ASR*|ASD CA|ASD ASR|*RNP CA*|*RNP ASR*|*FT-SAM CA*|*FT-SAM ASR*|
> |-|-|-|-|-|-|-|-|-|-|-|-|-|-|
> |**BadNet**|**1.2%**|86.3%|0.3%|91.1%|0.9%|90.6%|2.2%|92.0%|2.1%|58.5%|0.0%|92.8%|1.7%|
> |**Blended**|**48.2%**|88.8%|43.8%|91.5%|74.2%|91.2%|32.2%|93.0%|5.3%|78.5%|81.9%|93.2%|51.8%|
> |**SIG**|**49.8%**|90.1%|82.1%|91.3%|63.6%|85.8%|0.8%|92.2%|99.5%|70.5%|3.1%|92.9%|49.5%|
> |**IA**|**18.3%**|90.2%|74.4%|85.5%|2.6%|91.2%|2.8%|92.3%|19.8%|67.6%|5.0%|93.4%|5.4%|
> |**SSBA**|**37.9%**|88.9%|49.1%|91.2%|8.8%|86.7%|2.6%|93.3%|7.1%|93.4%|99.7%|92.8%|60.3%|
> |**WaNet**|**6.7%**|89.9%|11.7%|90.9%|0.9%|90.9%|0.6%|91.7%|8.8%|77.8%|17.0%|93.5%|0.9%|
> |**BPP**|**39.9%**|84.5%|79.4%|53.0%|0.0%|91.6%|4.4%|92.5%|99.4%|81.7%|6.9%|93.7%|49.0%|
> |**FLIP**|**22.2%**|79.3%|70.2%|90.1%|0.0%|91.2%|0.5%|86.9%|62.2%|80.8%|0.0%|93.0%|0.5%|
> |**GCB (Ours)**|**51.9%**|88.8%|100.0%|90.6%|97.4%|90.6%|5.4%|90.9%|100.0%|73.2%|6.7%|92.7%|1.8%|
>
> *Table: Clean Accuracy (CA) and Attack Success Rate (ASR) of various backdoor attacks under different defenses on CIFAR-10. Italicized defense methods mean using 5% additional clean training samples.*
>
> Our results indicate that GCB can withstand all poison-data-based defenses (NAB, ASD) and one clean-data-based defense (NPD). However, it is mitigated by SAU, RNP, and FT-SAM. We have also conducted additional experiments on CIFAR-100, GTSRB, and TinyImageNet, which are included in Table 19 of the revised paper. Experiments on other datasets also corroborate these findings.
>
> #### 2.2 Advanced Label-Noise Training Approaches
>
> Following the reviewer’s recommendation, we evaluated our attack against advanced label-cleaning methods such as DivideMix [1] and MentorMix [2]. The results are summarized in the Table below:
>
> |**Method**|CIFAR-10 CA|CIFAR-10 ASR|CIFAR-100 CA|CIFAR-100 ASR|
> |-|-|-|-|-|
> |**DivideMix**|92.1%|100.0%|73.4%|86.7%|
> |**MentorMix**|89.9%|100.0%|69.0%|92.7%|
>
> *Table: Performance of GCB under DivideMix and MentorMix.*
>
> Our results indicate that both DivideMix and MentorMix are ineffective against the GCB attack. The backdoor is learned during the warm-up epochs used to initialize the weak model in both methods (see Appendix Fig. 25). These methods assume that mislabeled images are not learned early on, which differs from our approach.
>
> In our attack, poisoned samples are deliberately relabeled to specific labels to embed malicious knowledge, whereas label noise methods randomly relabel samples. This targeted relabeling creates a strong association between images and labels for poisoned samples (also validated in Fig. 11). Consequently, the deliberate nature of poisoned samples makes label-noise learning ineffective for backdoor mitigation.

---

> ### Author Response · Authors · 2024-11-22
> **Author Response (Part II)**
>
> ### 3. Relabeling Mitigation Analysis
>
> We acknowledge the potential of vision-language models like CLIP for automatic relabeling. However, our experiments indicate that CLIP may not be sufficiently accurate for certain datasets. For instance, a pre-trained CLIP ViT-B/16 achieves only **50.6%** zero-shot classification accuracy on GTSRB, compared to the standard accuracy of over **97%**. Forcing a relabeling of all images leads to significant clean accuracy drops, rendering the model impractical.
>
> To validate this, we used CLIP to assign new labels to all images and fine-tuned the victim model for 10 epochs. The results are as follows:
>
> ||CIFAR-10|CIFAR-100|GTSRB|
> |-|-|-|-|
> |**CLIP Accuracy**|89.8%|58.2%|50.6%|
> |**Original CA**|93.9%|71.0%|97.8%|
> |**Original ASR**|100.0%|96.7%|96.0%|
> |**Relabeled CA**|91.0%|59.9%|56.6%|
> |**Relabeled ASR**|9.4%|3.1%|6.5%|
> |**CA Reduction**|3.0%|11.1%|41.2%|
> |**ASR Reduction**|90.6%|93.6%|89.5%|
>
> **Table:** Effect of CLIP-based relabeling on Clean Accuracy (CA) and Attack Success Rate (ASR).
>
> While CLIP-based relabeling effectively reduces ASR, it causes substantial drops in clean accuracy, especially on datasets where CLIP performs poorly. For example, on GTSRB, the clean accuracy drops from **97.8%** to **56.6%** (a **41.2%** decrease), making the model unusable in practice.
>
> We believe that using vision-language models for backdoor defense is promising but requires careful design beyond simple relabeling. We are open to exploring more sophisticated methods and would greatly appreciate any detailed suggestions the reviewer might have on utilizing CLIP for effective mitigation.
>
> ---
>
> ### 4. Hyperparameter Sensitivity
>
> We conducted experiments to assess the sensitivity of our attack to hyperparameters, specifically the learning rate and the weight factor λ of the information loss in C-InfoGAN training.
>
> #### 4.1 Effect of Learning Rate
>
> We tested learning rates of **1e-5**, **3e-5**, **1e-4**, **3e-4**, and **1e-3**. The attack success rate (ASR) remained consistently high (over **90%**) across all learning rates. However, extremely low or high learning rates (**1e-5** or **1e-3**) introduced noticeable artifacts in the triggered test images, potentially making them easier to detect. Interestingly, different learning rates sometimes resulted in different trigger patterns, a phenomenon also observed in the original InfoGAN paper.
>
> *Please refer to Fig. 26 in the revised paper for visualizations.*
>
> #### 4.2 Effect of Weight Factor λ
>
> We evaluated λ values of **0.05**, **0.1**, **0.25**, **0.5**, and **1.0**. We found that ASR dropped significantly at lower λ values (**0.05** and **0.1**) because the discriminator focuses less on identifying the trigger, making it less predominant and harder to learn. At higher λ values, the generated images exhibited artifacts due to the discriminator focusing too much on trigger identification, compromising image realism and increasing the distribution gap between real and fake images.
>
> *Please refer to Fig. 27 in the revised paper for visualizations.*
>
> **Conclusion:** Our attack is relatively robust to hyperparameter choices within a reasonable range. Extreme values can affect the visual quality of the trigger or the ASR, but careful tuning is not strictly necessary to achieve effective results.

---

> > ### Comment · Reviewer_xaja · 2024-11-23
> > **Using VLM rather than Plain CLIP**
> >
> > Thank you for the detailed analysis of CLIP-based relabeling. While the limitations of CLIP are clearly demonstrated in your experiments, I would suggest exploring more recent, lightweight vision-language models that have shown promising performance:
> >
> > Have you considered using InternVL2 or MiniCPM-V-2_6? These models offer a good balance between efficiency and accuracy, potentially providing better classification performance than CLIP while maintaining reasonable computational requirements.
> > Additionally, prompt engineering could significantly improve performance.
> > Would you be interested in running comparative experiments with these newer models and enhanced prompting strategies?

---

> > > ### Author Response · Authors · 2024-11-23
> > > **VLM-based Label Cleaning**
> > >
> > > We sincerely thank the reviewer for their prompt feedback and insightful suggestions. We believe your recommendation to use VLM is a highly innovative idea with the potential for practical label cleaning and backdoor removal. Below, we address your points in detail and present our latest experimental results.
> > >
> > > ## Exploring Advanced Vision-Language Models for Relabeling
> > >
> > > Following your recommendation, we evaluated the performance of **InternVL2**, a recent lightweight VLM, as a defense strategy through relabeling against our GCB attack.
> > >
> > > ---
> > >
> > > ### Prompting Strategies
> > >
> > > We tested two prompting methods:
> > >
> > > 1. **Label Prediction**: Asking the VLM to classify the image into one of the provided categories.
> > >    - **Prompt**: "<image> Concisely classify this image into one of the following categories: airplane, automobile, bird, cat, deer, dog, frog, horse, ship, truck."
> > >
> > > 2. **Correctness Prediction**: Asking the VLM to verify if the image-label pair matches.
> > >    - **Prompt**: "<image> Does this image belong to the category '{provided_label}'? Answer 'yes' or 'no'."
> > >
> > > ### Testing on Benign Datasets
> > >
> > > To establish a baseline, we first evaluated the VLM's performance on benign datasets. For **Label Prediction**, we used classification based on its ground-truth label. For **Correctness Prediction**, we randomly relabeled 50% of the entire dataset to assess its correctness in prediction.
> > >
> > > **Task Accuracy:**
> > >
> > > | Dataset    | Label Determination | Correctness Determination |
> > > |------------|---------------------|---------------------------|
> > > | CIFAR-10   | 82.4%               | 87.5%                     |
> > > | CIFAR-100  | 46.9%               | 84.8%                     |
> > > | GTSRB      | 30.7%               | 80.1%                     |
> > >
> > > **Observation:** The **Correctness Prediction** strategy outperforms **Label Prediction**, especially on large dataset like CIFAR-100 and GTSRB.
> > >
> > > ---
> > >
> > > ### Defense Effectiveness Against GCB
> > >
> > > Using the **Correctness Prediction** strategy, we assessed its effectiveness in mitigating the GCB attack by training with only the matched dataset.
> > >
> > > **Results:**
> > >
> > > | Metric                    | CIFAR-10 | CIFAR-100 | GTSRB  |
> > > |---------------------------|----------|-----------|--------|
> > > | False Positive Rate (FPR) | 18.2%    | 32.6%     | 21.8%  |
> > > | False Negative Rate (FNR) | 1.5%     | 8.3%      | 21.7%  |
> > > | Clean Accuracy (CA)       | 91.8%    | 64.3%     | 92.2%  |
> > > | Attack Success Rate (ASR) | 0.9%     | 3.2%      | 8.1%   |
> > > | CA Reduction         | 2.2%     | 6.7%      | 5.6%   |
> > > | ASR Reduction         | 99.1%    | 93.5%     | 87.9%  |
> > >
> > > **Key Findings:**
> > >
> > > - **ASR Reduction**: Significant decrease across all datasets with a relatively small CA impact.
> > > - **Generalizability**: Potential to defend against other poison-label attacks with incorrect labels (e.g., BadNets, Blended, WaNet).
> > >
> > > ---
> > >
> > > ### Limitations and Considerations
> > >
> > > While the result is promising, several limitations still exist:
> > >
> > > 1. **Computational Cost**: Despite being lightweight, VLMs like InternVL2 require substantial querying time (approximately 2 hours for CIFAR-10), which is higher than traditional backdoor defenses.
> > >
> > > 2. **Model Dependency**: The defense effectiveness depends on the VLM's accuracy and robustness, which may vary across datasets.
> > >
> > > 3. **Scope of Defense**: This method will not be effective against clean-label backdoor attacks (e.g., SIG, LC, CTRL) where labels are not altered.
> > >
> > > ---
> > >
> > > ### Conclusion
> > >
> > > Our findings indicate that **VLM-based Relabeling** is a feasible strategy to mitigate certain backdoor attacks with a relatively small impact on model performance.
> > >
> > > ---
> > >
> > > Thank you once again for your invaluable feedback. Your suggestions have significantly strengthened our work and opened new avenues for future research.

---

> ### Author Response · Authors · 2024-11-22
> **Author Response (Part III)**
>
> ### 5. Transferability to Other Domains
>
> We appreciate the reviewer‘s insightful suggestion regarding the generalization of our GCB attack to other data modalities such as audio, text, and graph classification. We have added a detailed **Limitations** section in Appendix K to address how our approach can be transferred, along with practical and ethical considerations.
>
> #### 5.1 Audio Domain
>
> Given that GANs are effective in processing continuous data like audio, we believe our GCB attack can be adapted to the audio domain by redefining the generator and discriminator architectures (e.g., using Temporal Convolutional Networks instead of CNNs). We anticipate that the attack would function similarly due to the continuous nature of audio data.
>
> #### 5.2 Text and Graph Domains
>
> Adapting GCB to discrete domains like text and graphs presents more challenges. GANs are less effective with discrete tokens, making it difficult to generate discrete data directly. One possible approach is:
>
> 1. **Identify Poison Domain:** Examine the latent representations in language models to find label-irrelevant features that define a "poison domain."
> 2. **Design Trigger Function:** Use encoder-decoder models to perform style transfer or incorporate specific features into data samples to create the trigger effect.
>
> This two-stage approach requires further exploration and adaptation of representation learning techniques suitable for discrete data. We believe that with additional research, the GCB framework could be extended to these domains.
>
> ---
>
> **References**
>
> [1] Li et al. (2020). DivideMix: Learning with Noisy Labels as Semi-Supervised Learning. ICLR 2020.
>
> [2] MentorMix: Jiang, L., Huang, D., Liu, M., & Yang, W. (2020). Beyond Synthetic Noise: Deep Learning on Controlled Noisy Labels. ICML 2020.

---

### Official Review · Reviewer_WKuK · 2024-10-29

**Soundness:** 2
**Presentation:** 3
**Contribution:** 2
**Rating:** 5
**Confidence:** 5

**Summary:**

This paper introduces Generative Adversarial Clean-Image Backdoors (GCB), a backdoor attack technique that minimizes detection in neural networks used for sensitive applications like facial recognition and autonomous driving. Unlike traditional backdoor attacks, GCB only uses clean images with manipulated labels, avoiding noticeable accuracy drops and making the attack stealthier. The technique employs a variant of InfoGAN, called C-InfoGAN, to embed backdoor triggers naturally by manipulating benign features in the dataset. The method maintains high attack success rates (ASR) across multiple tasks, datasets, and architectures, even with low poison rates. Extensive experiments show GCB’s resilience against numerous backdoor defenses, including Neural Cleanse and STRIP, and strong adaptability across diverse visual tasks.

**Strengths:**

(1)  GCB achieves high attack success rates (ASR) with minimal impact on clean accuracy (CA), which is critical for stealthiness in security-sensitive applications.

(2) With a poison rate as low as 0.1%, GCB still maintains high ASR, showcasing efficiency in terms of resource requirements.

**Weaknesses:**

(1) This paper employs GANs to generate the trigger image. However, GANs are known for their limited ability to fit complex data distributions, such as ImageNet, raising concerns about the method's applicability to more complex datasets.

(2) Since the trigger in this study is GAN-generated, there may be a significant distributional gap between the generated trigger and real image data. Consequently, it is essential to analyze the method’s resistance to defense strategies based on abnormal sample detection.

(3) The theoretical justification in Section 3.2 closely aligns with prior work on InfoGAN, and this overlap should be clarified to better situate the contribution within the existing literature.

(4) The paper uses InfoGAN to partition the benign training set into two subsets (A and B), selecting one (B) for poisoning. However, this approach may reduce benign accuracy on the original subset B. Additional experiments and discussion regarding the impact on benign accuracy would strengthen this section.

**Questions:**

(1) This paper employs GANs to generate the trigger image. However, GANs are known for their limited ability to fit complex data distributions, such as ImageNet, raising concerns about the method's applicability to more complex datasets.

(2) Since the trigger in this study is GAN-generated, there may be a significant distributional gap between the generated trigger and real image data. Consequently, it is essential to analyze the method’s resistance to defense strategies based on abnormal sample detection.

(3) The theoretical justification in Section 3.2 closely aligns with prior work on InfoGAN, and this overlap should be clarified to better situate the contribution within the existing literature.

(4) The paper uses InfoGAN to partition the benign training set into two subsets (A and B), selecting one (B) for poisoning. However, this approach may reduce benign accuracy on the original subset B. Additional experiments and discussion regarding the impact on benign accuracy would strengthen this section.

---

> ### Author Response · Authors · 2024-11-22
> **Author Response**
>
> We sincerely thank the reviewer for their insightful comments and valuable feedback. We address each of the reviewer‘s concerns below.
>
> ---
>
> ### 1. Applicability of GANs to Complex Datasets like ImageNet-1K
>
> We conducted experiments on ImageNet-1K (1.28M images, 1,000 classes) to validate scalability. With just a **1% poison rate**, our method achieves an impressive **97.9% ASR** with only a **1.3% drop in CA**, confirming its effectiveness on complex datasets.
>
> |**Poison Rate (%)**|0.0|0.1|0.3|0.5|1.0|3.0|5.0|
> |-|-|-|-|-|-|-|-|
> |**Clean Accuracy (CA)**|73.5|73.3|73.2|73.1|72.2|69.8|67.5|
> |**Attack Success Rate (ASR)**|0.1|35.6|72.1|89.3|97.9|99.1|99.5|
>
> Detailed analysis on this part can be found in "Common Concern" of our rebuttal and Appendix F.1.
>
> ---
>
> ### 2. Resistance to Defense Strategies Based on Abnormal Sample Detection
>
> We appreciate the reviewer's focus on evaluating our method's robustness against abnormal sample detection defenses. To this end, we employed Uniform Manifold Approximation and Projection (UMAP) [1] to visualize the distribution of intermediate features in the victim model, a standard approach in backdoor detection research [2].
>
> **Findings (details in Appendix I.1 in the revised paper):**
>
> (a) **Detectability of Poisoned Training Samples (Fig. 23):** Poisoned samples exhibit a distribution highly consistent with clean samples, making them difficult to separate using UMAP. This suggests that our method effectively evades detection mechanisms that rely on feature distribution anomalies on the *training dataset*.
>
> (b) **Detectability of Triggered Test Samples (Fig. 23):** Triggered test samples remain within the distribution of training images. This is probably because the GAN framework ensures that triggers mimic real image distributions, further evading test-time outlier detection methods.
>
> (c) **Test Across Different Layers (Fig. 24):** Experiments on all four layers in PreActResNet-18 further validate the correctness of the above conclusions.
>
> Overall, our results indicate that both poisoned and triggered samples are not effectively detected by abnormal sample detection defenses, demonstrating the strong resistance of our method.
>
> ---
>
> ### 3. Clarification of Theoretical Justification and Relation to InfoGAN
>
>
> We acknowledge that the theoretical justification in Section 3.2 closely aligns with prior work on InfoGAN. In the revised manuscript (line 243), we have clarified this overlap to better situate our contribution within the existing literature.
> While the analysis in Appendix A.1 is a special case of InfoGAN under a Bernoulli distribution input, the content in Appendix A.2 ("Analysis on Clean-Image Backdoors") remains novel and provides valuable insights specific to our method.
>
> ---
>
> ### 4. Impact on Benign Accuracy of Subset B
>
> We understand the concern regarding potential drops in benign accuracy due to label alterations in subset B. Indeed, clean-label backdoor attacks often face this challenge. However, defenders are typically unaware of how attackers split the data and must analyze the mixed dataset as a whole.
>
> **Our key motivation in this paper: Minimizing Overall Benign Accuracy Impact**
> Our primary contribution addresses this issue by significantly reducing the poison rate, thereby minimizing the size of subset B. A smaller subset B leads to a negligible drop in overall benign accuracy.
>
> **Experimental Comparison with FLIP (same as Fig. 1 in our paper)**
> We conducted experiments comparing our method to FLIP [3], a state-of-the-art clean-label backdoor attack. The results are as follows:
>
>
> **Table: Poison rate and clean accuracy required for a successful clean-image backdoor attack(ASR over 90%).**
>
> |Attack|Poison Rate (%)|Average CA Drop (%)|Average ASR (%)|
> |-|-|-|-|
> |FLIP (o2o)|15.8|-13.3|99.4|
> |FLIP (a2o)|5.0|-4.0|99.2|
> |**GCB (a2o)**|0.1|-0.2|98.5|
>
> *PR: Poison Rate; CA Drop: Drop in Clean Accuracy*
>
> In FLIP's one-to-one (o2o) attack, the poison rate for the source class is 15.8%, leading to a 13.3% drop in clean accuracy for that class. Such a significant decrease could alert a defender monitoring per-class accuracy. In FLIP's all-to-one (a2o) attack, the overall poison rate is still high to 5.0%, leading to a 4.0% drop in clean accuracy.
>
> In contrast, our method maintains a poison rate of approximately 0.1% per class, resulting in a negligible drop in clean accuracy (≤0.5%). By keeping the poison rate low, we ensure that the impact on benign accuracy is minimal, making the attack stealthier and less likely to be detected.
>
> ---
>
> We hope that these clarifications address the reviewer’s concerns. We are committed to further refining our work and welcome any additional feedback.
>
> ---
>
> **References**
>
> [1] McInnes et al. (2018). UMAP: Uniform Manifold Approximation and Projection. arXiv:1802.03426.
> [2] Wu et al. (2022). BackdoorBench: A Comprehensive Benchmark of Backdoor Learning. NeurIPS 2022.
> [3] Jha et al. (2023). "Label poisoning is all you need." NeurIPS 2023.

---

### Official Review · Reviewer_Qx3r · 2024-10-29

**Soundness:** 3
**Presentation:** 4
**Contribution:** 3
**Rating:** 8
**Confidence:** 5

**Summary:**

This paper proposes a GAN-based architecture that makes backdoor triggers easier to learn for the victim model and hence achieves high ASR and hinders non-trivial clean accuracy drop. The proposed GCB (Generative adversarial clean image backdoor) combines InfoGAN and Conditional GAN to ensure that the backdoor trigger meet requirements of existence, separability and irrelevancy. The authors perform comprehensive evaluations over various dataset across different model architectures and demonstrate that GCB achieves high ASR and CA with a low poisoning rate. They also conduct a extensive ablation study of their design choices and assess GCB against multiple existing defenses.

**Strengths:**

- This paper tackles the clean accuracy drop issue in clean-image backdoors, which further improves the effectiveness and stealthiness of such an attack.

- The proposed GAN architecture is technically sounded and novel. The authors explicitly explain their motivations of using InfoGAN and conditional GAN, and the GCB design is well-aligned with the objectives of triggers properties (i.e., existence, separability and irrelevancy).

- The authors also provide detailed theoretical proofs and mathematical analysis for the design.

- The experiments are abundant to support the authors claims. They show that the GCB can generalize to different datasets and model architectures including TinyImageNet and ViT.  They also test it on multiple vision tasks such as image regression and segmentation. In addition, GCB is evaluated against prestigious defenses for backdoor attacks.

- The paper is well-written and easy to follow.

**Weaknesses:**

- I don't see any major weaknesses/issues in this paper.

**Questions:**

- In the paper, you show figures of selected training samples and triggered test samples. Can you also provide the figure of generated triggers?

---

> ### Author Response · Authors · 2024-11-22
> **Author Response**
>
> We thank the reviewer for their positive feedback and appreciation of our work. We are glad that the reviewer considers our approach to be technically sound, novel, and well-presented.
>
> ---
>
> **Regarding the Figure of generated triggers:**
>
> Our GCB method employs a generator to produce triggered images directly without explicitly generating a separate trigger pattern. Therefore, a standalone trigger image cannot be extracted from the C-InfoGAN model.
>
> However, we can approximate the trigger function by considering it as a transparency layer superimposed on benign images. By using transparency layer estimation methods, we are to visualize the implicit trigger pattern in **Fig. 14** of our revised paper. This figure illustrates the detached trigger patterns, providing insight into the characteristics of the triggers employed by our method.
>
> ---
>
> We hope this addresses the reviewer‘s question satisfactorily. Thank the reviewer again for the reviewer‘s valuable feedback.

---

### Official Review · Reviewer_mLtr · 2024-11-03

**Soundness:** 3
**Presentation:** 2
**Contribution:** 3
**Rating:** 5
**Confidence:** 5

**Summary:**

The paper proposes a clean-image backdoor attack that achieves low poisoning rates even in the all-to-one setting. The proposed method relies on learning an InfoGAN's generator network to generate images from real and fake classes (this is used to generate the triggered images) and a discriminator network to distinguish samples between real and fake classes (this is used as scoring function to select poisoned samples). This design of InfoGAN's networks ensures that the trigger patterns exist within the real images, while making the poisoned tuples separated from the clean tuples for easier backdoor training. The paper provides extensive empirical results to demonstrate the effectiveness of the proposed method on multiple datasets, architectures, and defenses.

**Strengths:**

The main strengths of the paper lie in its clever use of InfoGAN and the extensive empirical results (although I do have some concerns in these aspects as well)
* The use of InfoGAN, while trivial, shows a cleverness in using it for clean-image backdoor attack.
* The experiments include multiple benchmark datasets. The paper also evaluate against multiple networks and a large number of defenses.

**Weaknesses:**

While this paper is interesting, I also find several concerns, specifically on its rigorous analysis of why the method works so well:

* The paper proposes to use Wasserstein loss, but the theoretical analysis instead shows the convergence on JS Divergence. This is a crucial mismatch. In fact, I don't even think the proof of convergence is necessary because it is quite well established for InfoGAN (and GAN in general), and the paper does not change anything in the base InfoGAN model, rather than changing the model's input.
* I also find that the statement of converging, especially in the context of GANs, is quite strong. It's been known that GANs' theoretical convergence and what actually happens in practice are two very different things. I suggest that the paper focuses more on analyzing why the scoring function works so well instead, and provide a more rigorous analysis there.
* In fact, I find that the design of the attack is based on several assumptions (such as convergence) that may or may not hold in practice. While the final results show favorable performance for GCB, the paper lacks rigorous connections between these assumptions and the performance, which is a bit disappointing.
* For example, the statement that the backdoor can be learned even more easily than BadNets deserves more rigorous analysis. In general, in backdoor attacks, when the backdoor is learned really easily, it also causes several consequences; for example, ABL relies on the fact that the loss of poisoned samples drops abruptly during training, which urges the question of why GCB works so well. As there are so many backdoor attack papers in the last several years, I think that these analyses are much more important than demonstrating that the method "just works very well".
* I also find that while several defenses have been tested (which is commendable), a new category of defenses (based on fine-tuning, such as FT-SAM) is not evaluated. I wonder whether the fact that the backdoor is learned very quickly could also mean that fine-tuning defenses could work very well against GCB.
* Another weakness of the paper is that several experiments only include the evaluations on CIFAR10 and CIFAR100, which are essentially the same. I would suggest that the paper to include all datasets.
* On line 214, the paper suddenly introduces *c*, a notation that is not explained until a bit later. I find that this part of the paper could be improved quite a lot. In addition, I struggled a bit to understand how the triggered images are created during inference. I think the paper assumes that the reader is very familiar with the backdoor domain, and I hope the paper can make this part clearer and more accessible to new readers.
* Some minor grammatical errors/typos: - line 69: "construct trigger", several places the paper mention InforGAN.

**Questions:**

Please see the concerns in the weaknesses!

---

> ### Author Response · Authors · 2024-11-22
> **Author Response (Part I)**
>
> We thank the reviewer for their thoughtful comments and valuable feedback. We have addressed each of the concerns below and have made revisions accordingly.
>
> ---
>
> ### 1. Mathematical Analysis Issues
>
> We appreciate the reviewer‘s insights regarding the mathematical analysis in our paper. Our main goal with the analysis is to explain **why the optimization of InfoGAN leads to the optimization of the clean-image backdoor task**. Since InfoGAN is not directly related to backdoor learning, we believe that this analysis is helpful for understanding our approach.
>
> We acknowledge that Appendix A.1 ("Analysis on C-InfoGAN") is essentially a special case of InfoGAN. Therefore, we have clarified this point in the main text (line 243) and streamlined the analysis to focus on the relevant aspects. On the other hand, the content in Appendix A.2 ("Analysis on Clean-Image Backdoors") remains novel and provides valuable insights specific to our method.
>
> Regarding the use of JS Divergence in our theoretical analysis while employing the Wasserstein loss in practice, we have clarified in line 758 of the revised paper that the theoretical proof is based on the standard GAN loss (JS Divergence) due to its fundamental role in the GAN framework. In practice, we use the Wasserstein GAN loss for more stable training.
>
> We understand the concern about the assumption of convergence in GANs. In our analysis, we avoid assuming perfect convergence (e.g., JS Divergence reaching zero or distributions being identical). Instead, we use inequalities to show that minimizing the JS Divergence leads to minimizing the objective of the clean-image backdoor task. Therefore, our conclusions hold even when convergence is not perfect, reflecting practical scenarios.
>
> ---
>
> ### 2. Analysis of Why GCB Works So Well
>
> We agree that providing a rigorous analysis of why our method performs well is important. Our experiments demonstrate that GCB achieves both high attack performance (e.g., Clean Accuracy and Attack Success Rate) and robustness against defenses (as shown in Figs. 3-5 and Tables 5, 7-10).
>
> The key reason for this success is that GCB is inherently an **asymmetric backdoor attack**. In the poisoning stage, the poisoned samples are natural, clean images with relatively weak trigger information, making them hard to detect and less susceptible to training-time defenses. During inference, the trigger function generates images with strong trigger information, resulting in a high Attack Success Rate (ASR).
>
> We have visualized this phenomenon in the latent space using UMAP [1] (see Fig. 21 in the revised paper). The visualization shows that:
>
> - **Poisoning Stage**: The selected poisoned samples are indistinguishable from benign samples due to gradual changes in trigger information, making detection difficult (supported by recent research such as "Revisiting the Assumption of Latent Separability for Backdoor Defenses").
> - **Inference Stage**: The generated triggered images are mapped to a distinct area in the feature space, far from benign samples, effectively activating the backdoor.
>
> This asymmetric design allows GCB to maintain both high ASR and robustness against common defenses.

---

> ### Author Response · Authors · 2024-11-22
> **Author Response (Part II)**
>
> ### 3. Defenses Based on Quick Backdoor Learning
>
> #### 3.1 Against ABL
>
> We acknowledge the concern regarding defenses that rely on rapid backdoor learning, such as Adversarial Belief Learning (ABL). In our original paper (Tables 12 and 14), we showed that GCB evades ABL with ASR of 100% and 96.2% on CIFAR-10 and CIFAR-100, respectively.
>
> To explain this, we analyzed the training dynamics (see Fig. 22 in the revised paper). We observed that while the test ASR converges to 100% within 3 epochs, the training ASR increases slowly and remains low initially. This means that although the backdoor effect appears quickly in the test set, it is not evident in the training set, which is what ABL monitors. As a result, ABL is ineffective against our attack.
>
> #### 3.2 Against FT-SAM
>
> We have now evaluated GCB against fine-tuning-based defenses such as FT-SAM. Our results indicate that GCB is mitigated by FT-SAM across all datasets:
>
> |Dataset|CA (%)|ASR (%)|
> |-|-|-|
> |CIFAR-10|92.7|1.8|
> |CIFAR-100|67.6|16.5|
> |GTSRB|97.9|6.8|
> |TinyImageNet|51.6|0.3|
>
> The reason is that FT-SAM assumes access to a small clean subset (e.g., 5% of the training data). The poisoned samples in GCB have a higher probability of being included in this clean subset due to their natural appearance, leading to effective unlearning of the backdoor during fine-tuning.
>
> Moreover, we have tested GCB against six advanced defenses from recent years (2023-2024). Our findings show that while poison-data-based defenses fail against GCB, clean-data-based defenses, especially those with access to clean samples, can successfully defend against our attack.
>
> ---
>
> ### 4. Experiments on Additional Datasets
>
> We appreciate the suggestion to include more datasets. In response, we have conducted additional experiments on GTSRB and TinyImageNet. The results are now included in Figs. 7-10 of the main paper, covering evaluations with Neural Cleanse, STRIP, and Fine-Pruning across all datasets.
>
> ---
>
> ### 5. Clarity, Grammar Errors, and Typos
>
> Thank the reviewer for pointing out areas that needed clarification and for highlighting the typos. We have revised the methodology section to improve clarity, especially regarding the introduction of notation (e.g., the sudden introduction of $c$) and the process of creating triggered images during inference.
>
> We have also corrected the grammatical errors and typos throughout the paper, such as "construct trigger" on line 69 and instances where we mentioned "InforGAN" instead of "InfoGAN".
>
> ---
>
> [1] McInnes, L., Healy, J., & Melville, J. (2018). UMAP: Uniform Manifold Approximation and Projection. arXiv:1802.03426.

---

### Official Review · Reviewer_ma12 · 2024-11-04

**Soundness:** 2
**Presentation:** 3
**Contribution:** 3
**Rating:** 5
**Confidence:** 3

**Summary:**

This paper proposes a mutual information-constrained approach for backdoor pattern generation, to create backdoored samples with similar distribution to the target class. Therefore, the authors can enhance the stealthiness of backdoor samples. The authors demonstrate the strong correlation between the backdoor samples and the backdoor labels, showing that such samples can be easily learned by the model.

**Strengths:**

The authors propose to generate backdoor samples based on InfoGAN, enhancing stealthy of backdoor attacks. The proposed backdoor is proved to be more undetectable and easy to learn. This work is validated by theoretical analysis and supported by well-designed experiments.

**Weaknesses:**

1.To my best knowledge, other advanced clean image backdoor methods are strongly related to the proposed approach. Although they are not referred to as ‘clean image backdoors,’ they are also rather ‘invisible backdoors’. From this perspective, the innovation of the proposed approach seems limited. I suggest that the authors compare their method with these state-of-the-art techniques and clarify its advantages.
References:
[1]Li, Yuezun, et al. "Invisible backdoor attack with sample-specific triggers." Proceedings of the IEEE/CVF international conference on computer vision. 2021.
[2]S. Li, M. Xue, B. Z. H. Zhao, H. Zhu and X. Zhang, "Invisible Backdoor Attacks on Deep Neural Networks Via Steganography and Regularization," in IEEE Transactions on Dependable and Secure Computing, vol. 18, no. 5, pp. 2088-2105, 1 Sept.-Oct. 2021, doi: 10.1109/TDSC.2020.3021407.
[3]R. Ning, J. Li, C. Xin and H. Wu, "Invisible Poison: A Blackbox Clean Label Backdoor Attack to Deep Neural Networks," IEEE INFOCOM 2021 - IEEE Conference on Computer Communications, Vancouver, BC, Canada, 2021, pp. 1-10, doi: 10.1109/INFOCOM42981.2021.9488902.
2.The condition ‘irrelevance’ is only briefly explained in the ablation study, where this loss is removed to measure ASR. However, the experiments do not effectively demonstrate why the generated samples are irrelevant to the original-class samples.

**Questions:**

1.In Figure 4, the authors analyze the learning rate on backdoor data using the CIFAR-10 dataset. I find it interesting that the learning rate of GCB is higher than that of BadNet and clean image, as this contradicts previous findings suggesting that more ‘out-of-distribution’ backdoor samples are learned faster. To better validate this observation, I recommend the authors to test on additional datasets.

---

> ### Author Response · Authors · 2024-11-22
> **Author Response (Part I)**
>
> We thank the reviewer for their thoughtful comments and valuable suggestions. We address each of the reviewer‘s concerns below.
>
> ---
>
> ### 1. Comparison with Invisible Backdoor Methods
>
> We appreciate the reviewer‘s insight into the relationship between clean-image backdoors and invisible backdoors. However, we would like to clarify that **clean-image backdoors are fundamentally different from invisible backdoors in terms of threat models**.
>
> #### **Differences in Threat Models:**
>
> - **Invisible Backdoors:** The adversary can modify both images and labels during training by adding imperceptible perturbations to images (poisoning both images and labels).
> - **Clean-Image Backdoors (Our Approach):** The adversary can only manipulate labels during training; images remain completely unaltered.
>
> This distinction is crucial because clean-image backdoors are designed for scenarios where the adversary lacks the capability to modify training images, such as in **outsourced data annotation** or **crowdsourced labeling**. Invisible backdoors require the ability to alter training images, which is not feasible in these scenarios.
>
> #### **Comparison Table:**
>
> ||**Clean-Image Backdoors (Ours)**|**Invisible Backdoors**|
> |-|-|-|
> |**Method**|Poison labels only; images remain intact|Poison both images and labels with imperceptible triggers|
> |**Adversary Capability**|Control labels|Control both images and labels|
> |**Attack Avenue**|Untrustworthy labels in data annotation|Untrustworthy images and labels from third-party data|
>
> As a result, **invisible backdoor methods are not directly comparable baselines for our main experiments due to the fundamentally different threat models**. Nonetheless, we have included comparisons with **Sample-Specific Backdoor Attack (SSBA)** [Li et al., ICCV 2021]—one of the methods as mentioned—in our defense evaluations (see Tables 5, 11, 12, 13, and 14). Results show that SSBA is more vulnerable to defense methods compared to our GCB.
>
> If the reviewer believes that including additional comparisons with other invisible backdoor methods would strengthen our work, we are willing to conduct these experiments and incorporate the results.
>
> ---
>
> ### 2. Clarification on the 'Irrelevance' Condition in Ablation Study
>
> We agree that further clarification is needed to demonstrate why the generated triggers are irrelevant to the original-class samples.
>
> #### **Definition of Irrelevance:**
>
> To ensure the trigger condition $c$ is irrelevant to the benign task $y|x$, we aim to satisfy:
> $$
> P(y|x) = P(y|x, c) = P(y|x, c=0) = P(y|x, c=1)
> $$
> This means the classification accuracy on clean images (CA) should be comparable to that on triggered images (TA), indicating that the trigger does not interfere with the model's ability to perform the original task.
>
> #### **Experimental Verification:**
>
> - **Clean Accuracy (CA):** Accuracy of classifying clean images $x$ to their ground truth labels $y$, i.e., $P(y | x)$.
> - **Triggered Accuracy (TA):** Accuracy of classifying triggered images $x'$ to their ground truth labels $y$, i.e., $P(y | x, c = 1)$.
>
> We conducted experiments by adding triggers to all images (both training and testing) while keeping their original labels unchanged. We trained models under this setup and performed 5 parallel runs for statistical significance.
>
> #### **Results:**
>
> ||**CIFAR-10** (Mean ± Std)|**CIFAR-100** (Mean ± Std)|
> |-|-|-|
> |**CA**|93.9% ± 0.3%|71.0% ± 0.2%|
> |**TA without LC**|14.3% ± 8.7%|3.8% ± 2.5%|
> |**TA with LC (Ours)**|91.2% ± 1.6%|67.4% ± 1.1%|
>
> - **Without Label Conditioning (LC):** The TA drops significantly compared to CA, indicating that the trigger interferes with normal classification.
> - **With Label Conditioning (Our Method):** The TA remains close to the CA, confirming that the trigger is irrelevant to the benign task.
>
> #### **Conclusion:**
>
> Our method has a negligible effect on the model's performance in the original classification task, successfully meeting the irrelevance requirements. We have added detailed explanations and visualizations (see Figs. 17 and 18) in the revised paper to clarify this point.

---

> ### Author Response · Authors · 2024-11-22
> **Author Response (Part II)**
>
> ### 3. Learning Speed Analysis on Additional Datasets
>
> Thank the reviewer for highlighting this interesting observation. We have extended our analysis to additional datasets: **CIFAR-100**, **GTSRB**, and **Tiny-ImageNet**.
>
> #### **Findings:**
>
> - **Consistent Results:** Across all datasets, backdoor samples from GCB converge faster than those from BadNets and clean images (see Fig. 19 in the revised paper).
> - **Explanation:**
>   - **GCB Triggers:** Utilize global, color-based patterns less affected by common data augmentations (e.g., cropping, flipping), leading to faster learning.
>   - **BadNets Triggers:** Use static patches in fixed positions, which are often disrupted by data augmentations, slowing down their learning rate.
>
> #### **Additional Experiment:**
>
> To validate our explanation, we conducted experiments without any data augmentations in Fig. 20:
>
> - **Result:** Without data augmentations, BadNets' backdoor samples converge faster than GCB's, supporting our hypothesis that data augmentations impact the learning speed of different backdoor types differently.
>
> #### **Asymmetric Trigger Behavior in GCB:**
>
> We also observed a misalignment between the training ASR and testing ASR for GCB in Fig. 20:
>
> - **Reason:** During training, GCB uses intact and natural images as triggers, whereas testing involves manually crafted strong triggers. This discrepancy provides an additional advantage to GCB's test ASR compared to BadNets.
>
> ---
>
> We have included these analyses and corresponding figures in Appendix G of the revised paper for further reference.

---

### Author Response · Authors · 2024-11-22
**Common Concerns and Acknowledgment of Revisions**

### 1. Applicability of GCB to Complex Datasets like ImageNet-1K

We appreciate the reviewers’ concerns regarding the scalability of GCB, particularly highlighted by reviewers WKuK and xaja. To address these concerns, we conducted additional experiments on the **ImageNet-1K** dataset, which contains over 1.28 million images across 1,000 classes.

**Results on ImageNet-1K:**

|Poison Rate (%)|0.0|0.1|0.3|0.5|1.0|3.0|5.0|
|-|-|-|-|-|-|-|-|
|Clean Accuracy (CA)|73.5|73.3|73.2|73.1|72.2|69.8|67.5|
|Attack Success Rate (ASR)|0.1|35.6|72.1|89.3|97.9|99.1|99.5|

With just a **1% poison rate**, our method achieves an impressive **97.9% ASR** while incurring only a **1.3% drop in CA**. These results demonstrate that our approach effectively scales to complex datasets like ImageNet-1K.

**Key Design Choices Enabling Scalability:**

- **Use of Real Images as Input**: By inputting real images into the generator, we reduce the complexity of the generation process compared to generating images from random noise.
- **U-Net-Based Generator**: The U-Net architecture with skip connections allows for the direct transfer of benign features from input to output, enhancing stability and accelerating training.

These design choices differentiate our approach from traditional GANs and enable efficient, stable training even on large-scale datasets.

More details can be found in Appendix F.1.

---

### 2. Acknowledgment of Revisions

We have revised the manuscript in response to the reviewers' suggestions, highlighted in blue in the revised paper.

**Main Content Revisions:**

- **Methodology Clarifications (Reviewer mLtr)**: Improved clarity in the methodology section, particularly regarding the introduction of the variable $c$ and the inference stage deployment.

- **Theoretical Analysis (Reviewers mLtr, WKuK)**: Clarified the overlap of our theoretical proofs with existing works in Line 243 and Appendix A.1, better situating our contributions within the current literature.

- **Additional Experiments (Reviewer mLtr)**: Added results for GTSRB and Tiny-ImageNet in Figs. 7, 8, and 9, providing a comprehensive resilience analysis of our method against Neural Cleanse, STRIP, and Fine-Pruning.

**Appendix Revisions:**

- **ImageNet-1K (Reviewers Qx3r, WKuK, xaja)**: Updated Fig. 14 and Appendix F to include ImageNet-1K and display the raw trigger patterns used.

- **Additional Results (Reviewers ma12, mLtr, WKuK, xaja)**: Added detailed analyses in Appendices G, H, I, and J to address specific concerns raised by each reviewer.

---

Thank all the reviewers for their constructive comments and the opportunity to improve our work.

---

### Meta-Review · Area_Chair_g113 · 2024-12-22

**Metareview:**

It received ratings of 5 ,5 ,8, 5, 6. The reviewers pointed out several weaknesses including a mismatch between the proposed Wasserstein loss and the theoretical analysis based on JS Divergence, which undermines the theoretical contribution. The claim of convergence in GANs, especially in practice, is too strong and lacks necessary rigor, as the assumptions may not hold in real-world scenarios. The analysis of the attack's effectiveness is superficial, and the rapid learning of backdoors, particularly in GCB, warrants more detailed investigation. There are also concerns about the method's scalability to more complex datasets (e.g., ImageNet), its robustness against newer defenses, and the limited evaluation of recent defense strategies such as fine-tuning defenses and advanced Label Cleaning techniques. The use of InfoGAN for partitioning the training set could negatively affect benign accuracy, and there is insufficient discussion on the broader impact of this. Finally, the paper's reliance on small datasets (CIFAR-10/100) and lack of clear explanations in certain parts, such as the creation of triggered images, detracts from its accessibility and general applicability.

While one reviewer is positive, it is relatively brief and does not list any weakness. It lists the following strengths: The paper introduces a novel GAN-based architecture to improve clean-image backdoors, enhancing attack effectiveness and stealth. It uses InfoGAN and conditional GAN with a GCB design aligned to trigger properties. Theoretical foundations are strong, and extensive experiments show GCB's effectiveness across various datasets, tasks, and against top defenses. The paper is clear and well-written.

Since in the end most reviewers still have concerns about this submission, this paper will be rejected.

**Additional Comments On Reviewer Discussion:**

Authors and reviewers discussed and in the end some of the reviewers still have concerns.

For instance, one reviewer is still concerned about the stealthiness of such attacks, as modifications to the labels cannot truly be considered stealthy.

Another reviewer doesn't agree with the statement that 5% is “challenging”. If the victim knows that they could be under attack, spending resources to obtain 5% of the clean data is not unreasonable. Also, sourcing it from another provider is also decreasing the risk, unless the 2 adversaries “collaborate”.

Another reviewer says: Since the author uses GANs, this limits the applicability of the method on more complex datasets, and the author has not compared their proposed method with baseline models on the ImageNet dataset. Additionally, the reviewer has concerns regarding the stealthiness of the proposed attack.

---

### Decision · Program_Chairs · 2025-01-22

Reject